# 🎣 FISHER: A FOUNDATION MODEL FOR MULTI-MODAL INDUSTRIAL SIGNAL COMPREHENSIVE REPRESENTATION

## ABSTRACT

Industrial signal analysis has emerged as a critical problem for the industry. Due to severe heterogeneity within industrial signals, which we summarize as the M5 problem, previous works could only deal with small sub-problems by training specialized models, which lacks robustness and incurs huge burdens during development and deployment. However, we argue that the M5 problem can be dealt by scaling up, where dealing with the multi-sampling-rate is the first step. In this paper, we propose FISHER, a Foundation model for multi-modal Industrial Signal compreHEnsive Representation. To support arbitrary sampling rates, FISHER considers the increment of sampling rate as the concatenation of sub-band information. Specifically, FISHER takes the STFT sub-band as the modeling unit and adopts a teacher-student SSL framework for pre-training. To evaluate the model performance, we also develop the RMIS benchmark, which consists of 19 datasets across four modalities. FISHER is compared with 15 SOTA speech/audio/music encoders, demonstrating versatile and outstanding capabilities with a general performance gain of at least 3.23%. Meanwhile, FISHER possesses much more efficient scaling curves, where even FISHER-tiny with 5.5M parameters outperforms huge baseline encoders up to 2B. We further reveal that the key to success is adaptively utilizing the full signal bandwidth regardless of the sampling rate. Both FISHER and RMIS will be open-sourced.

## 1 INTRODUCTION

Recent years saw rapid deployment of supervisory control and data acquisition (SCADA) systems in modern manufacturing. These SCADA systems employ ubiquitous sensors of various modalities to continuously monitor and analyze the production lines, generating a huge volume of streaming industrial signals round-the-clock. Nowadays, the installation of SCADA systems do not present any major technical challenge. However, how to efficiently analyze these signals and accurately detect malfunctions are critical challenges for the industries, due to the unique heterogeneity of industrial signals. In this paper, we boil it down to the **M5** problem:

- **Multi-modal**. Sound, vibration, voltage, current, temperature, etc.
- **Multi-sampling-rate**. The sampling rate is often selected as twice the Nyquist bandwidth to reduce cost. Common sampling rates range from 3 kHz to 50 kHz.
- **Multi-scale**. Due to the differences in operating mechanisms (sliding, rotation, static, etc) and working conditions, the signal characteristics are diverse.
- **Multitask**. Anomaly detection, fault diagnosis, remaining useful life (RUL) estimation, etc.
- **Minim fault**. Fault data are often scarce, and the class distribution is often imbalanced.

Compared with speech data, audio data, and music data, industrial signal data are not scarce. However, due to the M5 problem, large-scale pre-training has rarely been explored for industrial signal. Previous works mainly focus on small sub-problems, such as sound-based anomaly detection (Jiang et al., 2023; 2024; Wilkinghoff, 2024), vibration-based bearing fault diagnosis (Wang et al., 2023; Peng et al., 2025), and vibration-based RUL estimation (Wang et al., 2018). These works usually

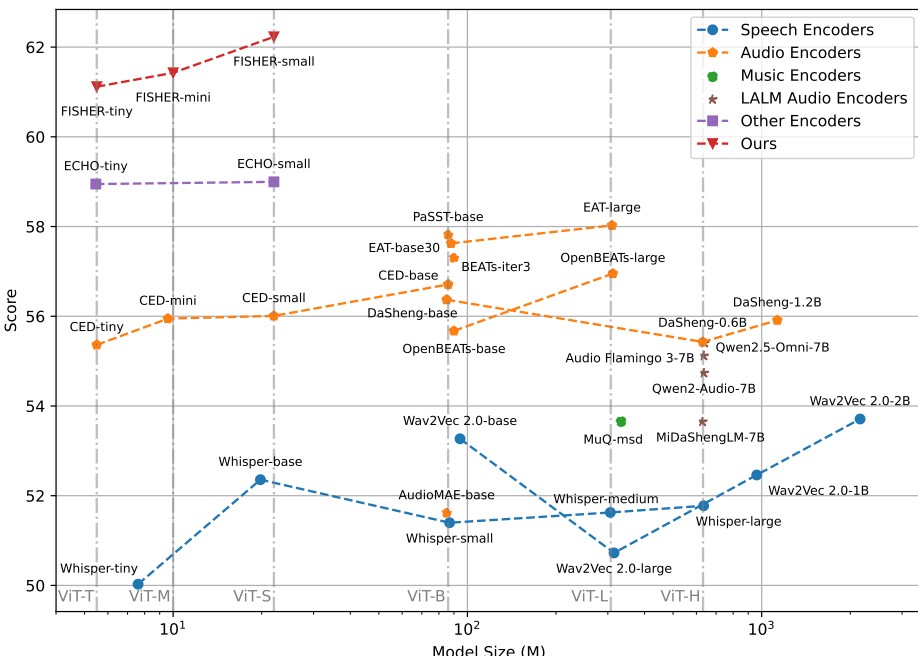

Figure 1: Model Performances on the RMIS benchmark, where the higher the score is, the better the model is. Compared with top baselines, FISHER achieves superior performances with much smaller model sizes, demonstrating versatile capabilities and efficient scaling properties.

train specialized models on small-scale datasets, resulting in models being deficient in robustness under diverse working conditions. Moreover, it incurs huge burdens in the development and deployment of SCADA systems, since each sub-problem must be dealt by an exclusive model.

Therefore, we aim to develop a universal and powerful signal encoder for heterogeneous industrial signals, in order to 1) significantly improve the quality of the signal representation and 2) greatly reduce the complexity of developing and deploying SCADA systems. Recent advances in vision foundation model (Siméoni et al., 2025), large language model (LLM) (Xu et al., 2025; Guo et al., 2025), and large audio language model (LALM) (Goel et al., 2025; Dinkel et al., 2025) have shown that large-scale pre-training can mitigate external heterogeneity, uncover internal similarities and thereby building up powerful foundation model that generalizes well on various tasks. As for industrial signals, we argue that their heterogeneity mainly lies in appearance, while there are still some internal similarities that have yet to be explored, which are listed as follows:

- Sound and vibration, the two most common modalities, are essentially different observations of vibration, since sound is recorded by the oscillation of the microphone diaphragm.
- Different signals are perceptions of the same mechanical event by different physical laws.
- Machines are assembled from components. Signals of different machines are comparable.

Thus, we conjecture that it is also viable to overcome the M5 problem and build up foundation models for industrial signals via large-scale pre-training. Specifically, the multi-modal, multi-scale, and multitask problems can be gradually alleviated by scaling up (Kaplan et al., 2020), while the minim fault problem can be dealt by the external knowledge injected during pre-training (Jiang et al., 2024; Zheng et al., 2024). The multi-sampling-rate problem, however, is a crucial and unavoidable problem for scaling up pre-training, since information is sparsely encoded in the full bandwidth of the signal. Almost all speech/audio pre-trained models only accept inputs with a fixed sampling rate, and data of higher sampling rates must be resampled. This incurs a huge loss of high frequency information, which is crucial for industrial signals as demonstrated in Section 5.5. As a solution, the model must be able to cope with arbitrary sampling rates in order to leverage the full bandwidth.

In this work, we propose **FISHER**, short for **F**oundation model for multi-modal **I**ndustrial **S**ignal compre**HE**nsive **R**epresentation. As the first work in our series, FISHER mainly deals with the

unavoidable multi-sampling-rate problem. As known, higher sampling rates incorporate more information about the signal, thus FISHER models the increment of sampling rate as the concatenation of additional sub-band information. Specifically, the raw signal, regardless of its modality, is represented by short time Fourier transform (STFT) with fixed-duration window and hop size. The spectrogram is then split into sub-bands with predefined bandwidth and the model processes these sub-bands individually. The model is trained by a teacher-student self distillation framework (Chen et al., 2024), where the student is guided by the representations of the teacher, and the teacher is an exponential moving average (EMA) version of the student.

To comprehensively evaluate the model, we also develop the **RMIS** benchmark, short for **R**epresentation of **M**5 **I**ndustrial **S**ignals. The RMIS benchmark incorporates 19 sub-datasets with two typical signal analysis tasks, i.e. anomaly detection (no fault as prior) and fault diagnosis (classify specific fault type). All models are evaluated by k-nearest neighbor (KNN) inference to demonstrate the inherent capabilities.

We compare FISHER with top speech/audio/music foundation models and LALM audio encoders, where FISHER showcases versatile performances and efficient scaling properties. FISHER achieves an overall score of 62.23% on the RMIS benchmark, surpassing all baselines by 3.23%. Meanwhile, FISHER possesses a much more efficient scaling curve, achieving superior performances with much smaller sizes. We further demonstrate that the performance gain is attributed to its ability to adaptively utilize the full signal bandwidth, whereas all baselines can only utilize a portion of it. Our main contributions are:

- We demonstrate for the first time that it is feasible to train unified foundation models for industrial signals that generalize on multiple signal analysis tasks across modalities.

- We propose FISHER, a **F**oundation model for **I**ndustrial **S**ignal compre**HE**nsive **R**epresentation. FISHER models the STFT sub-bands via teacher-student self-distillation, which enables it to process arbitrary industrial signals without resampling.

- We propose the RMIS benchmark to evaluate the inherent abilities on signal analysis tasks, where we compare top speech/audio/music encoders and LALM audio encoders.

- FISHER achieves superior performances on the RMIS benchmark with much smaller model sizes, showcasing more efficient scaling properties.

## 2 RELATED WORKS

Industrial signals are continuous one-dimensional series, which means that relevant experiences from speech and audio can be leveraged. Recent years saw huge advancements in speech encoders, such as the Wav2Vec 2.0 series (Baevski et al., 2020), WavLM (Chen et al., 2022) and the Whisper series (Radford et al., 2023). Compared with speech, general audio is sparser, lacks annotations, and more closely resembles industrial signals. Top audio encoders conformably adopt SSL pre-training, where the most popular paradigm is the MAE framework (He et al., 2022). Typical works are AudioMAE (Huang et al., 2022) and DaSheng (Dinkel et al., 2024b). Another choice is the teacher-student self-distillation framework in Data2Vec 2.0 (Baevski et al., 2023), CED (Dinkel et al., 2024a) and EAT (Chen et al., 2024). Iterative self-tokenization and prediction is also effective, such as BEATs (Chen et al., 2023b) and MuQ (Zhu et al., 2025). With the rise of LALM, these models are utilized as the audio encoder to LLM, such as Whisper in Xu et al. (2025); Ding et al. (2025), BEATs in Tang et al. (2023) and DaSheng in Dinkel et al. (2025).

On the other hand, pre-trained models are revolutionizing the field of industrial signal analysis with the advantage of versatility. In anomalous sound detection (ASD), fine-tuning audio pre-trained models has become the dominant approach (Jiang et al., 2024; Lv et al., 2024). For vibration-based fault diagnosis, transferring image pre-trained backbones was once the research hotspot (Wen et al., 2020; Li et al., 2022) since images and industrial signals are both sparse. Some recent works employ LLM to directly model the signal series (Zhang et al., 2022; Pang et al., 2024). BearLLM (Peng et al., 2025) builds up a LALM-like model for bearing fault analysis with a vibration encoder pre-trained on vibration data. However, since common faults in fault diagnosis are supported by considerable data volume, they usually have in-depth theoretical analysis such as characteristic frequency. Thus, mechanism-aware small models are still competitive for fault diagnosis (Chen et al., 2023a).

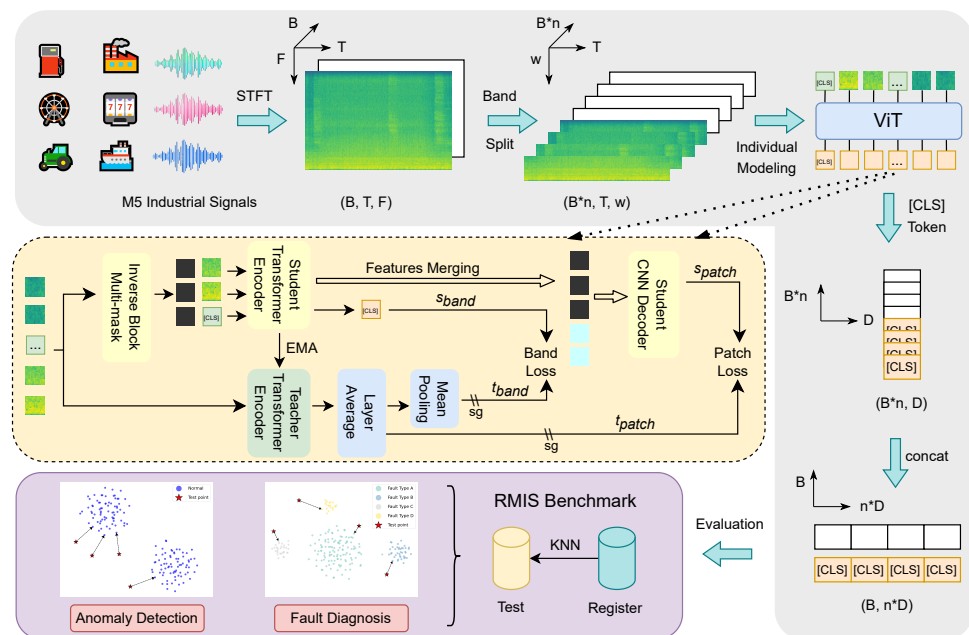

Figure 2: Pipeline of FISHER and RMIS. FISHER converts signals into STFT spectrograms and splits them into sub-bands with fixed bandwidth $w$. These sub-bands are processed individually by the ViT backbone and the [CLS] embeddings are concatenated as the signal representations. The ViT backbone is trained by teacher-student self-distillation, where the teacher encoder is an EMA version of the student encoder. The model is evaluated on the RMIS benchmark by KNN inference.

# 3 FISHER

## 3.1 SUB-BAND MODELING

In FISHER, the input signal is first converted to STFT. While most audio pre-trained models adopt the log-mel spectrogram as the input representation, FISHER reverts to STFT since:

- Malfunctions often appear in high frequencies, which would be diluted in mel scale.
- The harmonic relationships of frequencies are vital, which would be smoothed in mel scale.

To deal with the multi-sampling-rate problem, the STFT window size $N$ is mapped to fixed time duration $t_{win}$. That is, let $f_s$ denote the signal sampling rate, then $N = t_{win} \cdot f_s$. In this way, the frequency resolution of STFT will be constant for arbitrary sampling rates:

$$\Delta f = \frac{f_s}{N} = \frac{f_s}{t_{win} \cdot f_s} = \frac{1}{t_{win}} \tag{1}$$

where $\Delta f$ is the frequency gap between adjacent frequency grid. Similarly, the STFT hop size is also mapped to fixed time duration $t_{hop}$, such that signals with the same time duration will have spectrograms with the same time shape regardless of the sampling rate.

To deal with the variable frequency shape, FISHER emphasizes the importance of sub-band and considers it as the building blocks of the overall information. On the one hand, as depicted in Figure 3, the information gain of a higher sampling rate lies in the additional sub-bands. As known, all sensors employ anti-aliasing filtering to prevent signal aliasing. Therefore, the spectrogram does not contain any information about frequencies higher than half the sampling rate. On the other hand, sampling rates of common large-scale datasets, i.e. 16 kHz, 32 kHz, 44.1 kHz and 48 kHz, are integer multiples of a fundamental frequency $f_{base}$, such as 2 kHz and 4 kHz, making sub-band a natural unit for modeling multi-sampling-rate signals. Thus, we take the sub-band as the unit

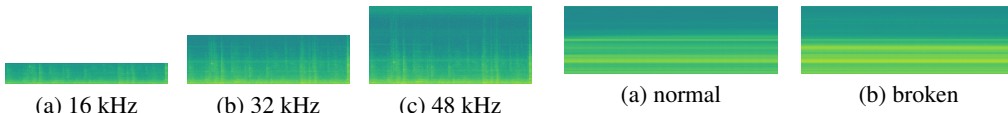

(a) 16 kHz      (b) 32 kHz      (c) 48 kHz         (a) normal      (b) broken

Figure 3: STFT Spectrograms of the same source under different sampling rates. Here we adopt fixed-duration window and hop size. A higher sampling rate comprises additional high frequency sub-bands that carry extra information, while its low frequency sub-bands are almost identical with that of a lower sampling rate. Thus, it is heuristic to select the sub-band as the modeling unit.

Figure 4: STFT Spectrograms of two vibration signals from the WTPG dataset. Both are extremely stationary throughout the entire clip (more than 300 s), causing the split segments to be highly identical. If random splitting were adopted, the detection results would grow extremely high (up to 99%), making it impossible for comparison.

for modeling, and build up the information of the whole spectrogram by concatenating sub-band information just like building blocks. That is, the signal representation is the concatenation of sub-band representations. The higher the sampling rate is, the more informative the representation is.

We now describe the sub-band modeling process in detail, which is illustrated in Figure 2. A batch of signals are first resampled to a batch-specific sampling rate $sr_{batch}$ to align the spectrogram shape for batching, where $sr_{batch}$ is randomly selected from all the harmonics of $f_{base}$ that are less than a maximum frequency $f_{max}$. The aligned signals are then converted to log amplitude STFT spectrograms of shape $(B, T, F)$, where $B$ is the batch size, and $T$ and $F$ are the time and frequency shapes respectively. Each spectrogram is then split into sub-bands with bandwidth $w$ and concatenated along the batch axis, thereby transferring the variability from the frequency axis to the batch axis. These sub-bands have a shape of $(B \times n, T, w)$, where $w = f_{base} \cdot t_{win}$ is the frequency gap on the spectrogram corresponding to $f_{base}$, and $n = \lfloor \frac{F}{w} \rfloor$ is the number of sub-bands. They are then processed individually by the network and their representations are concatenated afterwards.

## 3.2 NEURAL PROCESSING

FISHER adopts an encoder-decoder architecture with a teacher-student self distillation scheme for pre-training, which has been demonstrated effective in multiple self-supervised learning (SSL) models (Chen et al., 2024; Baevski et al., 2022; Siméoni et al., 2025). FISHER comprises three sub-networks: a student encoder $E_{stu}$, a student decoder $D_{stu}$ and a teacher encoder $E_{tea}$. Both encoders adopt the identical ViT (Dosovitskiy et al., 2021) structure with fixed sinusoidal position encoding and post-norm, while the parameters of $E_{tea}$ are the exponential moving average (EMA) of the parameters of $E_{stu}$:

$$\theta_{E_{tea}} = \tau \theta_{E_{tea}} + (1 - \tau) \theta_{E_{stu}} \tag{2}$$

where $\theta_{E_{tea}}$ and $\theta_{E_{stu}}$ are the parameters of $E_{tea}$ and $E_{stu}$ respectively, and $\tau$ is the EMA decay factor. $E_{tea}$ is updated per step. $D_{stu}$ is a shallow convolutional neural networks (CNN).

As illustrated in Figure 2, both encoders accept the STFT sub-band as the input, where it is further split into a patch sequence following the ViT style. For the student branch, the patch sequence is masked by inverse block masking adopted in EAT (Chen et al., 2024) with a large mask ratio of 80%, and masked patches are discarded. Here we follow the mask cloning strategy in EAT to efficiently increase the batch size, yet we constrain the maximum number of cloned sub-bands in a batch as $m_b$ to prevent fluctuation in GPU memory. The unmasked patches are then appended with a [CLS] token at front and sent into $E_{stu}$. After encoded by $E_{stu}$, the output of the [CLS] token is selected as the student sub-band representation $s_{band}$, and the output of these unmasked patches are merged with the masked parts in the original spatial locations, where the values of the masked parts are initialized from normal Gaussian. $D_{stu}$ takes in the merged sequence and outputs the student patch representation $s_{patch}$. For the teacher branch, $E_{tea}$ processes the unmasked patch sequence, and the embeddings of all its layers are averaged to derive the teacher sub-band representation $t_{band}$ and teacher patch representation $t_{patch}$. The self distillation process is supervised from both the

sub-band level and the patch level:

$$\begin{cases} L_{band} = \|s_{band} - sg(t_{band})\|_2^2 \\ L_{patch} = \|s_{patch} - sg(t_{patch})\|_2^2 \end{cases} \tag{3}$$

where $sg(\cdot)$ denotes stop gradient. The final loss is the combination of the two losses:

$$L = L_{band} + L_{patch} \tag{4}$$

It is noted that sub-bands are processed individually during training. During inference, only $E_{stu}$ is employed and its sub-band representations are concatenated to form the overall representation.

## 4 RMIS BENCHMARK

To evaluate the comprehensive representation capability of the model for M5 industrial signals, we develop the RMIS benchmark, which comprises six anomaly detection datasets and 13 fault diagnosis datasets, whose key features are presented in Table 1. To demonstrate the intrinsic versatility of the model, we evaluate the model by KNN inference without fine-tuning on any dataset. Each dataset produces a dataset score, which is based on either the area under the receiver operating characteristic curve (AUC) or accuracy, and we take the arithmetic mean of corresponding dataset scores as the task score. Finally, the overall benchmark score is the arithmetic mean of two task scores to eliminate the impact of data imbalance.

Table 1: Key features of datasets in the RMIS benchmark. Split denotes whether an official train-test split is provided for the dataset.

(a) Anomaly Detection

| Dataset | Modality | Num Class | Sampling Rate | Volume | Split |
|---------|----------|-----------|---------------|--------|-------|
| DCASE20 | Sound | 2 | 16 kHz | 153 h | ✓ |
| DCASE21 | Sound | 2 | 16 kHz | 165 h | ✓ |
| DCASE22 | Sound | 2 | 16 kHz | 139 h | ✓ |
| DCASE23 | Sound | 2 | 16 kHz | 50 h | ✓ |
| DCASE24 | Sound | 2 | 16 kHz | 49 h | ✓ |
| DCASE25 | Sound | 2 | 16 kHz | 45 h | ✓ |

(b) Fault Diagnosis

| Dataset | Modality | Num Class | Sampling Rate | Volume | Split |
|---------|----------|-----------|---------------|--------|-------|
| IICA | Sound | 6 | 48 kHz | 47 h | ✗ |
| IIEE | Sound | 3 | 44.1 kHz | 1 h | ✓ |
| WTPG | Vibration | 5 | 48 kHz | 14 h | ✗ |
| MaFaulDa_sound | Sound | 10 | 50 kHz | 3 h | ✗ |
| MaFaulDa_vib | Vibration | 10 | 50 kHz | 16 h | ✗ |
| SDUST_bearing | Vibration | 10 | 25.6 kHz | 25 h | ✗ |
| SDUST_gear | Vibration | 7 | 25.6 kHz | 17 h | ✗ |
| UMGED_sound | Sound | 11 | 51.2 kHz | 59 h | ✗ |
| UMGED_vib | Vibration | 11 | 51.2 kHz | 176 h | ✗ |
| UMGED_vol | Voltage | 11 | 51.2 kHz | 117 h | ✗ |
| UMGED_cur | Current | 11 | 51.2 kHz | 117 h | ✗ |
| PU_vib | Vibration | 3 | 64 kHz | 3 h | ✗ |
| PU_cur | Current | 3 | 64 kHz | 6 h | ✗ |

### 4.1 ANOMALY DETECTION

Anomaly detection is to predict whether a signal is normal or anomalous when no anomalies are provided for training, which emphasizes the scarcity of fault data. We evaluate the model on the datasets of the annual DCASE ASD challenge, including DCASE20 (Koizumi et al., 2019; Purohit et al., 2019; Koizumi et al., 2020), DCASE21 (Tanabe et al., 2021; Kawaguchi et al., 2021; Harada et al., 2021), DCASE22 (Dohi et al., 2022b;a), DCASE23 (Dohi et al., 2023; Harada et al., 2023b), DCASE24 (Nishida et al., 2024; Harada et al., 2023a) and DCASE25 (Nishida et al., 2025). We use the official split and evaluate the model by the challenge criteria, which is based on AUC. For each dataset, we report the harmonic mean over both the development and the evaluation subsets. All models adopt the identical KNN-based anomaly detection pipeline as AnoPatch (Jiang et al., 2024), which is introduced in Appendix A.1.1 to make the paper self-contained.

### 4.2 FAULT DIAGNOSIS

Fault diagnosis is to identify the specific fault type or health state of a signal with labeled data provided in advance. The fault diagnosis part in the RMIS benchmark is sourced from seven publicly available datasets: IDMT-ISA-COMPRESSED-AIR (IICA) (Johnson et al., 2020), IDMT-ISA-ELECTRIC-ENGINE (IIEE) (Grollmisch et al., 2019), the WT-planetary-gearbox-dataset (WTPG) (Liu et al., 2023), the Machinery Fault Dataset (MaFaulDa) (Ribeiro, 2016), the UM-GearEccDataset (UMGED) (Li et al., 2025), the SDUST dataset (Wang et al., 2024; Zhang et al., 2024; Han et al., 2023; 2022) and the PU Dataset (Lessmeier et al., 2016). Details of these datasets are presented in Appendix A.2. To reveal the modality specific performance, we first divide these data by the modality, resulting in 13 datasets. For each dataset, we flatten all multi-channel signals

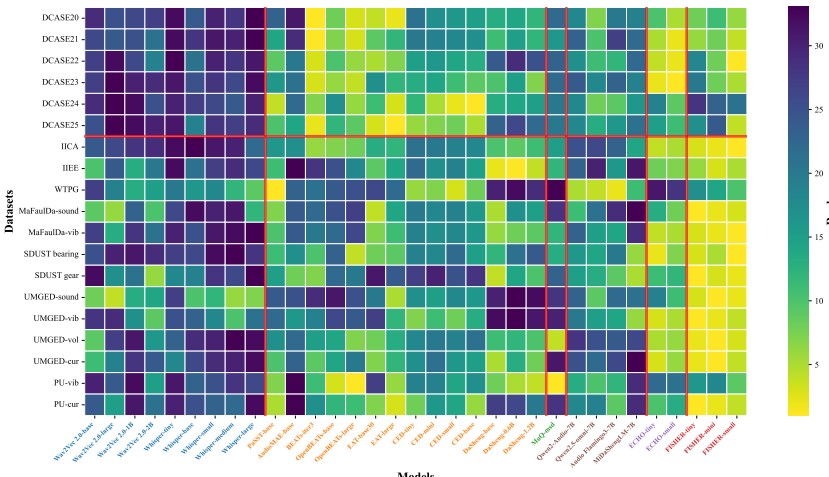

Figure 5: Ranking heatmap on the RMIS benchmark. We rank the scores of all models on each dataset. The brighter the color is, the better the model is. Audio encoders are distinguishably better than speech encoders, while LALM audio encoders are slightly better than speech encoders. FISHER demonstrates strong versatility on all datasets, especially on fault diagnosis tasks where signals are recorded in high sampling rates.

into single-channel and split them into segments if they are longer than 10 s. For KNN inference, $k$ is default to 5 for all models. All datasets are evaluated by macro-average accuracy.

To ensure proper difficulty of the task, the RMIS benchmark conduct sealed train-test split for datasets without official train-test split, where segments from the same channel of the same recording can not be partitioned into both the training and the test sets. Industrial signals are sometimes extremely stationary along the time axis due to the constant working condition, causing the split segments to be highly identical. If random splitting were adopted, these highly similar segments would appear in both the training and test sets, causing the task to be exceedingly simple. Therefore, we trace each channel of the original recording and allocate its segments into either the train set or the test set[1]. The train-test split ratio is default to 1:1, except for UMGED datasets which are 4:1. We ensure that all classes are presented in both sets and evaluate the model under 10 different splits. To eliminate the impact of fixed split ratio, the split ratio is further analyzed in Section 5.4.

# 5 EXPERIMENT

## 5.1 DETAILS OF FISHER

FISHER is trained under three scales, namely FISHER-tiny (5.5M), FISHER-mini (10M) and FISHER-small (22M), which are in line with the hierarchy of ViT. Table 2 lists unique hyperparameters for each version. As for the shared hyperparameters, $t_{win}$ and $t_{hop}$ are fixed as 25 ms and 10 ms, $f_{max}$ is 32 kHz. All models contain 12 layers and the patch size is $16 \times 16$.

Table 2: Unique Hyperparameters of FISHER

| Scale | Num Param | $f_{base}$ | $w$ | Hidden Size | Num Head | Batch Size[1] | $m_b$[1] | $m_l$[2] |
|-------|-----------|-----------|-----|-------------|----------|---------------|----------|----------|
| tiny  | 5.5M | 4000 | 100 | 192 | 3 | 24 | 64 | 5 |
| mini  | 10M  | 4000 | 100 | 256 | 4 | 32 | 64 | 5 |
| small | 22M  | 2000 | 50  | 384 | 6 | 16 | 128 | 2 |

[1] Both the batch size and $m_b$ are identified for each GPU.
[2] $m_l$ is the block size in inverse block masking.

All models are pre-trained on the combined dataset of Audioset (Gemmeke et al., 2017), Freesound[2], MTG-Jamendo (Bogdanov et al., 2019) and Music4all (Santana et al., 2020) with a total volume of 17k hours. We train each model for 400k steps on four NVIDIA RTX A6000 GPUs. For each model, we adopt a warm-up scheduler with a peak learning rate of 0.0005 and a warm-up step of 40k. During evaluation, FISHER directly processes the original signal without resampling.

---

[1]For the PU dataset, the signal under each working condition is recorded for 20 times, resulting in 20 highly identical segments. Thus, we allocate these segments based on the working condition.

[2]https://freesound.org/

Table 3: Results on the RMIS Benchmark (↑)

| Model | Variant | Anomaly Detection DCASE | | | | | | | IICA | IIEE | WTPG | MaFaulDa | | SDUST | | UMGED | | | | PU | | | All |
| | | 20 | 21 | 22 | 23 | 24 | 25 | Mean | | | | sound | vib | bearing | gear | sound | vib | vol | cur | vib | cur | Mean | |
|---|---|---|---|---|---|---|---|---|---|---|---|---|---|---|---|---|---|---|---|---|---|---|---|
| Wav2Vec 2.0 | base | 64.76 | 55.14 | 55.77 | 55.69 | 52.76 | 55.69 | 56.64 | 46.77 | 96.61 | 60.81 | 60.72 | 78.34 | 54.12 | 95.01 | 9.08 | 9.34 | 11.93 | 15.58 | 67.00 | 43.42 | 49.90 | 53.27 |
| | large | 65.61 | 55.23 | 54.44 | 48.09 | 50.51 | 33.98 | 51.31 | 45.49 | 72.48 | 78.33 | 62.83 | 87.35 | 49.98 | 97.04 | 10.12 | 8.40 | 9.18 | 12.50 | 69.24 | 48.86 | 50.14 | 50.72 |
| | 1B | 65.50 | 55.22 | 55.81 | 54.12 | 51.83 | 51.60 | 55.68 | 44.23 | 94.97 | 84.84 | 49.40 | 77.08 | 49.95 | 96.72 | 8.39 | 10.89 | 8.49 | 9.93 | 64.59 | 40.69 | 49.24 | 52.46 |
| | 2B | 65.49 | 55.47 | 56.35 | 55.47 | 53.91 | 54.53 | 56.87 | 42.09 | 78.81 | 84.49 | 59.96 | 84.14 | 50.31 | 98.24 | 8.37 | 12.06 | 10.80 | 11.13 | 72.67 | 44.07 | 50.55 | 53.71 |
| Whisper | tiny | 62.85 | 54.42 | 53.72 | 54.09 | 52.53 | 54.01 | 55.27 | 40.36 | 55.16 | 71.86 | 44.40 | 76.92 | 51.17 | 96.99 | 6.71 | 9.52 | 9.48 | 12.51 | 66.00 | 41.12 | 44.78 | 50.03 |
| | base | 65.66 | 54.89 | 56.12 | 56.90 | 53.70 | 56.14 | 57.24 | 40.10 | 77.24 | 83.95 | 36.35 | 80.82 | 51.30 | 96.90 | 8.98 | 9.71 | 9.02 | 10.21 | 69.95 | 42.70 | 47.48 | 52.36 |
| | small | 64.02 | 54.44 | 54.85 | 55.50 | 53.79 | 54.67 | 56.21 | 41.12 | 70.21 | 83.55 | 38.72 | 76.75 | 49.23 | 95.84 | 8.63 | 10.50 | 8.68 | 9.89 | 68.89 | 43.57 | 46.58 | 51.40 |
| | medium | 65.41 | 54.76 | 55.58 | 56.02 | 54.13 | 55.21 | 56.85 | 41.28 | 66.84 | 85.62 | 37.31 | 76.66 | 48.93 | 96.11 | 9.29 | 11.79 | 8.09 | 9.61 | 67.65 | 44.02 | 46.40 | 51.63 |
| | large | 62.27 | 53.65 | 54.51 | 53.97 | 52.53 | 54.54 | 55.25 | 47.57 | 70.44 | 87.24 | 59.32 | 73.36 | 50.85 | 94.08 | 9.14 | 10.10 | 8.34 | 8.05 | 68.80 | 40.68 | 48.31 | 51.78 |
| PaSST | base | 66.60 | 56.84 | 57.58 | 59.57 | 57.64 | 57.77 | 59.33 | 60.51 | 96.40 | 92.69 | 62.42 | 88.61 | 61.86 | 97.12 | 7.17 | 9.80 | 10.65 | 15.88 | 74.94 | 53.81 | 56.30 | 57.82 |
| AudioMAE | base | 63.87 | 54.86 | 56.34 | 57.96 | 55.30 | 56.82 | 57.52 | 58.88 | 53.89 | 64.02 | 47.48 | 80.95 | 59.74 | 98.18 | 6.91 | 10.93 | 10.01 | 11.59 | 56.02 | 35.54 | 45.70 | 51.61 |
| BEATs | iter3 | 74.20 | 60.50 | 58.95 | 63.31 | 56.93 | 59.09 | 62.16 | 77.93 | 68.17 | 70.57 | 52.87 | 84.45 | 62.37 | 98.23 | 5.98 | 10.32 | 10.95 | 17.23 | 74.18 | 48.42 | 52.44 | 57.30 |
| OpenBEATs | base | 70.86 | 58.60 | 57.45 | 62.48 | 56.10 | 57.32 | 60.47 | 77.87 | 70.93 | 63.60 | 45.50 | 83.39 | 57.31 | 96.51 | 5.47 | 9.51 | 11.09 | 13.78 | 76.96 | 49.50 | 50.88 | 55.67 |
| | large | 72.78 | 59.93 | 58.29 | 63.29 | 57.32 | 58.03 | 61.61 | 75.18 | 82.86 | 62.33 | 42.69 | 79.37 | 70.65 | 96.86 | 6.91 | 9.66 | 12.63 | 12.48 | 78.57 | 49.67 | 52.30 | 56.95 |
| EAT | base30 | 72.69 | 58.03 | 58.84 | 59.35 | 56.48 | 58.65 | 60.67 | 64.48 | 96.65 | 61.28 | 68.32 | 91.90 | 64.03 | 95.46 | 7.89 | 9.39 | 12.49 | 18.87 | 68.33 | 50.38 | 54.57 | 57.62 |
| | large | 73.81 | 57.06 | 57.94 | 60.29 | 57.78 | 60.34 | 61.20 | 65.80 | 94.06 | 68.91 | 58.47 | 88.42 | 62.64 | 96.32 | 9.57 | 11.60 | 11.22 | 15.54 | 75.74 | 54.73 | 54.85 | 58.03 |
| CED | tiny | 66.99 | 56.17 | 56.75 | 60.09 | 56.40 | 58.40 | 59.15 | 47.93 | 74.23 | 89.43 | 53.77 | 84.56 | 59.02 | 96.06 | 8.09 | 12.96 | 10.31 | 12.65 | 71.50 | 49.95 | 51.57 | 55.36 |
| | mini | 67.48 | 56.35 | 56.59 | 60.05 | 57.44 | 58.26 | 59.36 | 50.21 | 84.26 | 89.26 | 56.95 | 84.88 | 59.24 | 95.50 | 8.36 | 11.66 | 10.13 | 12.43 | 71.50 | 48.56 | 52.53 | 55.95 |
| | small | 67.50 | 56.65 | 56.87 | 60.76 | 57.88 | 58.15 | 59.63 | 50.49 | 80.23 | 91.20 | 54.67 | 84.67 | 58.00 | 96.12 | 8.05 | 12.15 | 10.26 | 13.14 | 72.89 | 49.08 | 52.38 | 56.01 |
| | base | 67.60 | 56.67 | 56.95 | 60.99 | 57.89 | 58.45 | 59.76 | 52.26 | 90.08 | 89.20 | 57.64 | 85.95 | 60.20 | 95.84 | 8.12 | 11.57 | 10.03 | 13.63 | 72.14 | 50.91 | 53.66 | 56.71 |
| DaSheng | base | 69.25 | 57.28 | 56.02 | 60.95 | 56.39 | 55.82 | 59.28 | 74.42 | 99.10 | 46.84 | 63.53 | 92.06 | 61.37 | 98.30 | 5.64 | 4.45 | 11.74 | 20.08 | 74.93 | 42.42 | 53.45 | 56.37 |
| | 0.6B | 68.35 | 56.76 | 55.35 | 59.85 | 56.34 | 55.41 | 58.68 | 75.14 | 99.25 | 45.41 | 57.26 | 91.11 | 58.23 | 97.59 | 5.27 | 4.28 | 11.69 | 14.68 | 75.78 | 42.63 | 52.18 | 55.43 |
| | 1.2B | 69.52 | 57.06 | 55.82 | 61.14 | 56.32 | 56.01 | 59.31 | 73.18 | 99.00 | 48.89 | 58.16 | 90.79 | 55.50 | 97.99 | 5.42 | 4.52 | 11.76 | 17.32 | 75.99 | 44.11 | 52.51 | 55.91 |
| MuQ | msd | 66.91 | 56.75 | 56.04 | 57.69 | 55.64 | 56.09 | 58.19 | 62.23 | 95.33 | 37.98 | 41.22 | 88.08 | 60.67 | 96.38 | 6.11 | 5.72 | 15.07 | 9.00 | 78.57 | 42.20 | 49.12 | 53.65 |
| Qwen2-Audio | 7B | 67.32 | 55.38 | 56.46 | 57.19 | 56.07 | 56.27 | 58.12 | 45.59 | 73.56 | 89.91 | 59.63 | 83.35 | 60.13 | 97.12 | 7.63 | 10.26 | 9.17 | 10.63 | 73.17 | 47.46 | 51.35 | 54.73 |
| Qwen2.5-Omni | 7B | 70.94 | 57.39 | 57.20 | 59.14 | 56.83 | 57.05 | 59.76 | 45.49 | 66.56 | 90.73 | 53.52 | 86.21 | 60.49 | 97.60 | 9.02 | 11.35 | 9.25 | 11.87 | 74.84 | 46.74 | 51.05 | 55.40 |
| Audio Flamingo 3 | 7B | 67.02 | 55.06 | 57.08 | 57.52 | 56.76 | 56.55 | 58.33 | 47.41 | 88.57 | 92.41 | 40.85 | 87.20 | 58.27 | 98.09 | 7.69 | 11.56 | 9.30 | 10.21 | 73.37 | 49.77 | 51.90 | 55.12 |
| MiDaShengLM | 7B | 67.10 | 55.46 | 56.10 | 58.70 | 56.15 | 56.03 | 58.26 | 63.78 | 59.63 | 86.39 | 35.69 | 76.94 | 68.31 | 98.25 | 7.75 | 15.54 | 9.20 | 7.29 | 67.47 | 41.28 | 49.04 | 53.65 |
| ECHO | tiny | 70.43 | 59.01 | 58.86 | 63.92 | 55.78 | 56.56 | 60.76 | 85.13 | 98.06 | 45.79 | 58.56 | 92.53 | 75.66 | 97.82 | 7.92 | 20.19 | 15.02 | 25.58 | 70.77 | 49.72 | 57.13 | 58.95 |
| | small | 72.50 | 60.20 | 59.13 | 64.25 | 56.76 | 57.45 | 61.72 | 84.74 | 98.11 | 50.75 | 61.52 | 92.11 | 67.76 | 97.63 | 8.85 | 17.90 | 12.91 | 19.09 | 70.86 | 49.37 | 56.28 | 59.00 |
| FISHER | tiny | 70.86 | 58.76 | 56.40 | 58.62 | 53.64 | 56.37 | 59.11 | 85.31 | 98.72 | 83.84 | 75.79 | 92.58 | 71.81 | 99.20 | 11.90 | 23.58 | 21.10 | 28.24 | 72.36 | 56.21 | 63.13 | 61.12 |
| | mini | 70.19 | 58.40 | 57.62 | 61.07 | 54.59 | 55.75 | 59.60 | 86.02 | 99.05 | 84.74 | 75.74 | 93.20 | 70.09 | 98.55 | 13.69 | 20.52 | 22.95 | 29.97 | 72.01 | 55.73 | 63.25 | 61.43 |
| | small | 71.04 | 59.48 | 59.64 | 62.63 | 55.62 | 58.46 | 61.15 | 90.23 | 98.96 | 86.77 | 72.61 | 95.86 | 76.35 | 99.08 | 12.72 | 18.35 | 17.42 | 25.52 | 74.90 | 54.29 | 63.31 | 62.23 |

Different colors denote speech encoders, audio encoders, music encoders, LALM audio encoders, other encoders and our model respectively.

## 5.2 BASELINES

We are curious about 1) how well current state-of-the-art (SOTA) speech/audio/music encoders can encode and understand industrial signals 2) pre-training on what kinds of data are effective for signal analysis 3) what kinds of pre-training/post-training are suitable for signal representation. Therefore, we select 15 SOTA encoders as baselines, including two speech encoders, seven audio encoders, one music encoder, four LALM audio encoders and an encoder inspired by FISHER. Details of these baselines are elaborated in Appendix A.3.

## 5.3 RESULTS ON THE RMIS BENCHMARK

Table 3 presents the numerical scores on the RMIS benchmark, where the higher the score is, the better the model is. Three conclusions about FISHER can be made. First of all, FISHER is the most versatile model for industrial signal analysis, which achieves a RMIS score of 62.23% and surpasses all baselines by at least 3.23%. FISHER is especially skilled for the fault diagnosis task, which incorporates large amount of high frequency signals. We will further demonstrate in Section 5.5 that the performance gain comes from adaptively utilizing the full signal bandwidth, while most baselines lose critical information during down-sampling. Secondly, FISHER demonstrates superior performances with much smaller model sizes. Three scales of FISHER with barely 5.5M, 10M and 22M parameters, outperform all baselines by at least 2.12%, 2.43% and 3.23%. Therefore, FISHER is more suitable for real-world deployment. Finally, FISHER is much more efficient for scaling on signal analysis tasks. As presented in Figure 1, the scaling curve of FISHER is constantly above the curves of all baselines. It is noted that large pre-trained models do not demonstrate dominant performance in industrial signal analysis as they have in speech and audio analysis, which suggests that models require tailored adjustments on industrial signals to enhance the analytical capabilities.

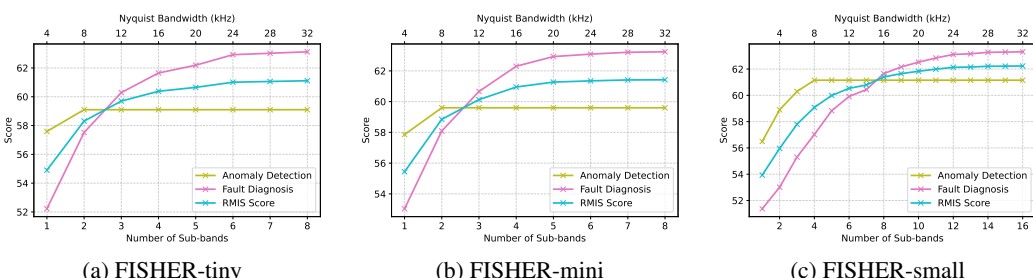

Figure 6: Performance of FISHER vs. Number of Available Sub-bands

Meanwhile, it is intriguing to notice some new findings on how to build signal foundation model by comparing these baselines. First of all, audio encoders are distinguishably better than speech encoders as depicted in Figure 5. It suggests that 1) speech data are probably not suitable for pre-training signal foundation model due the difference in inductive bias 2) network structure must be capable of dealing with sparse input. Secondly, LALM audio encoders are slightly better than speech encoders, indicating that additional post-training is beneficial for improving the analytical capabilities. Finally, as presented in Figure 1, the performances of most models continuously grow as the model size scales up, suggesting that scaling on signal analysis tasks is possible. However, the model requires specialized designs for industrial signal (such as how to deal with multi-sampling-rate) in order to further scale up, and the scaling law could be a bit different for industrial signal. We further discuss this part in Appendix A.4

### 5.4 MULTIPLE SPLIT RATIOS

In the RMIS benchmark, 12 out of 13 fault diagnosis datasets do not provide official split. Since previous works adopt various split ratios, we further analyze the model performance under multiple split ratios to eliminate its the impact. Specifically, we first plot the performance curve under variable split ratios ranging from 0.05 to 0.95 (train set ratio). For each split ratio, the model is still evaluated under 10 different train-test splits. Then we estimate the area under the multi-split curve. The results are presented in Table 5. There is a strong correlation between the score under fixed split ratio and the area under the multi-split-ratio curve, with a Pearson correlation of 0.967. Therefore, it is reasonable to evaluate the model under a fixed split ratio in the RMIS benchmark.

### 5.5 HIGH FREQUENCY GAIN

PaSST (Koutini et al., 2022), ECHO (Zhang et al., 2025b) and FISHER are the only models that accept input with sampling rates higher than 16 kHz. On the RMIS benchmark, these models all surpass their respective contemporary models, suggesting that high frequency components are crucial for analyzing industrial signals. To valid it, we constrain the number of sub-bands available to FISHER (starting from low frequency), and evaluate it on the RMIS benchmark. As depicted in Figure 6, the model performance grows steadily until it reaches the maximum sampling rate of the anomaly detection datasets. On fault diagnosis datasets with higher sampling rates, the performance continues to grow monotonically. This reveals that the success of FISHER is mainly attributed to the capability of to utilize the full bandwidth of the original signal.

## 6 CONCLUSION

In the paper, we hypothesized that the heterogeneous M5 industrial signals can be modeled in a unified model due to the intrinsic similarity. As a result, we proposed FISHER, which models the information gain of higher sampling rate as the concatenation of sub-band information. We also developed the RMIS benchmark to evaluate versatility on different signal analysis tasks, where FISHER excels all baselines by a wide margin with much efficient scaling properties. How to derive more powerful representation of industrial signal and effectively scale on downstream tasks will be the focus of our future work.

ETHICS STATEMENT

Our research promotes responsible AI development by advancing industrial safety and efficiency. The proposed FISHER model and RMIS benchmark are intended to improve anomaly detection and fault diagnosis, leading to more reliable machinery, safer working conditions, and the safeguarding of human health and well-being. Furthermore, the datasets used are publicly available and contain no commercially sensitive information. We are committed to fostering ethical research that serves the public good and contributes to a more sustainable future.

REPRODUCIBILITY STATEMENT

The SSL scheme, the model architecture and the training configurations are elaborated in the paper. The model checkpoints, the inference code of FISHER and the full evaluation pipeline of the RMIS benchmark will soon be open-sourced. The full training pipeline of FISHER will be open-sourced once accepted.

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

Table 4: Detailed Task Setup for Fault Diagnosis Datasets

| Dataset | Machine | Task | Classes |
|---|---|---|---|
| IICA | Air Compressor | Leakage | tubeleak_iO, tubeleak_niO, ventleak_iO, ventleak_niO, ventlow_iO, ventlow_niO |
| IIEE | Electric Engine | Fault | good, broken, heavy load |
| WTPG | Planetary Gearbox | Fault | broken, healthy, missing tooth, root crack, wear |
| MaFaulDa | Bearing | Fault | normal, horizontal misalignment, vertical misalignment, imbalance, underhang cage fault, underhang outer race, underhang ball fault, overhang cage fault, overhang outer race, overhang ball fault |
| SDUST_bearing | Bearing | Fault | NC, OF0.2, OF0.4, OF0.6, IF0.2, IF0.4, IF0.6, RF0.2, RF0.4, RF0.6, |
| SDUST_gear | Planetary Gearbox | Fault | NC, planetary fracture, planetary pitting, planetary wear, sun fracture, sun pitting, sun wear |
| UMGED | Gear | Eccentricity | E00, E02, E04, E06, E08, E10, E12, E14, E16, E18, E20 |
| PU | Bearing | Fault | healthy, IR, OR |

# A APPENDIX

## A.1 DETAILS OF THE RMIS BENCHMARK

### A.1.1 ANOMALY DETECTION

For ASD tasks in the RMIS benchmark, all models adopt the identical KNN-based anomaly detection pipeline as AnoPatch (Jiang et al., 2024) after extracting the embeddings, where normal embeddings from the training set form memory banks, and the anomaly score of each query embedding from the test set is the average distance to the nearest neighbors. The distance metric is selected as cosine distance and $k$ is kept as 1 for all datasets. To reveal the intrinsic capability of the model, we do not tune the hyperparameters of KNN on each ASD dataset. For the DCASE20 dataset, anomaly detection is conducted per machine id, where a memory bank is constructed for embeddings of each machine id, and each query embedding is inferred by the memory bank with the same machine id. For the rest DCASE datasets, anomaly detection is conducted per section. Since domain shift is involved, two memory banks are constructed for each section, one for the source embeddings and the other for the target embeddings.

## A.2 FAULT DIAGNOSIS

Table 4 presents the detailed task setup for fault diagnosis datasets, including detailed class mapping. For bearing fault diagnosis tasks, fault types are commonly similar, where NC, IF (some denote as IR), OF (some denote as OR), RF corresponds to normal condition, inner race fault, outer race fault and rolling element fault respectively.

## A.3 BASELINE DETAILS

Fifteen baselines are compared in this paper. We now describe the details of these baselines. For each baseline, we follow the official embedding procedure and select the top performing open-sourced checkpoints on the RMIS benchmark for fair comparison. We sort these baselines into 5 categories: speech encoders, audio encoders, music encoders, LALM audio encoders, and other encoders.

### A.3.1 SPEECH ENCODERS

We compare two classic speech pre-trained models: Wav2vec2.0 and Whisper.

For Wav2Vec 2.0 (Baevski et al., 2020), we utilize four official checkpoints: wav2vec2-base-960h (base), wav2vec2-x1s-r-300m (large), wav2vec2-xls-r-1b (1B), wav2vec2-x1s-r-2b (2B). During evaluation, we first extract the frame-level embeddings from the encoder and mean-pool them along the time axis to form the utterance-level embeddings.

For Whisper (Radford et al., 2023), we only utilize the encoder. We evaluate the encoders of five official pre-trained checkpoints (without fine-tuning): tiny, base, small, medium, and large-v3. Dur-

Table 5: Area under the Multi-Split Curve for Datasets without Official Split (↑)

| Model | Variant | IICA | WTPG | MaFaulDa | | SDUST | | UMGED | | | | PU | | Mean |
|---|---|---|---|---|---|---|---|---|---|---|---|---|---|---|
| | | | | sound | vib | bearing | gear | sound | vib | vol | cur | vib | cur | |
| Wav2Vec 2.0 | base | 46.05 | 57.32 | 57.17 | 73.94 | 52.37 | 89.69 | 10.67 | 11.76 | 12.21 | 14.97 | 65.28 | 41.99 | 44.45 |
| | large | 44.68 | 73.60 | 59.72 | 82.78 | 48.92 | 91.87 | 11.53 | 11.33 | 10.28 | 12.82 | 64.86 | 46.31 | 46.56 |
| | 1B | 43.51 | 80.46 | 46.58 | 73.11 | 48.46 | 91.12 | 10.93 | 13.45 | 9.90 | 10.81 | 64.03 | 41.54 | 44.49 |
| | 2B | 41.51 | 80.13 | 55.09 | 79.36 | 49.09 | 93.16 | 11.12 | 14.81 | 11.57 | 11.58 | 68.48 | 43.62 | 46.63 |
| Whisper | tiny | 39.87 | 67.38 | 43.29 | 72.69 | 49.75 | 91.80 | **9.27** | 12.23 | 10.47 | 12.48 | 63.85 | 40.95 | 42.84 |
| | base | 39.83 | 78.08 | 35.92 | 76.50 | 49.75 | 92.03 | 10.83 | 12.64 | 10.33 | 10.93 | 67.26 | 42.32 | 43.87 |
| | small | 40.60 | 77.57 | 38.43 | 72.55 | 47.55 | 90.70 | 10.47 | 12.92 | 10.03 | 10.60 | 66.15 | 43.03 | 43.38 |
| | medium | 40.97 | 80.61 | 37.01 | 72.56 | 47.90 | 91.14 | 10.78 | 14.18 | 9.80 | 10.59 | 64.76 | 43.00 | 43.61 |
| | large | 43.15 | 82.17 | 31.86 | 71.82 | 47.32 | 88.95 | 11.18 | 12.64 | 10.05 | 9.55 | 66.34 | 40.56 | 42.57 |
| PaSST | base | 58.23 | **87.20** | 58.88 | 84.11 | 59.74 | 92.75 | 9.77 | 12.60 | 10.96 | 14.29 | 70.56 | 51.71 | 50.90 |
| AudioMAE | base | 55.61 | 62.53 | 45.48 | 76.71 | 57.89 | 93.14 | 8.85 | 13.15 | 10.84 | 11.65 | 54.43 | 36.00 | 43.86 |
| BEATs | iter3 | 72.62 | 67.79 | 49.88 | 80.19 | 60.05 | 93.11 | 8.77 | 13.00 | 11.38 | 15.01 | 70.75 | 45.77 | 49.03 |
| OpenBEATs | base | 72.31 | 60.57 | 43.63 | 79.43 | 55.34 | 90.25 | 8.16 | 12.25 | 11.50 | 12.84 | 73.39 | 47.88 | 47.30 |
| | large | 70.11 | 61.13 | 40.86 | 75.05 | 67.36 | 91.07 | 9.36 | 12.39 | 13.16 | 12.16 | **74.98** | 47.97 | 47.97 |
| EAT | base30 | 61.49 | 58.89 | 64.81 | 87.88 | 61.14 | 89.46 | 9.71 | 11.17 | 12.22 | 16.21 | 65.27 | 48.91 | 48.93 |
| | large | 62.58 | 63.60 | 55.69 | 84.28 | 60.43 | 90.58 | 10.78 | 13.70 | 11.32 | 13.88 | 71.54 | 52.23 | 49.22 |
| CED | tiny | 47.03 | 84.02 | 50.13 | 80.38 | 57.11 | 91.01 | 9.46 | 14.88 | 10.87 | 12.19 | 67.57 | 48.00 | 47.72 |
| | mini | 48.75 | 84.72 | 53.95 | 80.65 | 57.06 | 90.62 | 9.50 | 13.78 | 10.86 | 12.06 | 67.74 | 46.84 | 48.04 |
| | small | 49.08 | 86.04 | 51.06 | 80.75 | 56.54 | 91.26 | 9.35 | 14.16 | 10.98 | 12.64 | 68.14 | 48.15 | 48.18 |
| | base | 50.53 | 84.32 | 53.75 | 81.71 | 58.38 | 91.07 | 9.22 | 13.68 | 10.88 | 12.71 | 68.06 | 48.90 | 48.60 |
| DaSheng | base | 70.01 | 47.67 | 58.49 | 88.66 | 59.10 | 92.21 | 7.94 | 6.60 | 11.67 | 16.86 | 71.70 | 41.67 | 47.72 |
| | 0.6B | 70.68 | 46.71 | 53.59 | 87.44 | 56.53 | 91.60 | 7.89 | 6.59 | 11.75 | 13.23 | 72.60 | 41.98 | 46.72 |
| | 1.2B | 68.75 | 49.56 | 54.12 | 86.66 | 53.85 | 91.58 | 8.11 | 6.68 | 11.61 | 14.99 | 72.63 | 43.10 | 46.80 |
| MuQ | msd | 59.08 | 38.25 | 40.35 | 84.51 | 59.48 | 89.97 | 8.81 | 8.56 | 14.91 | 9.89 | 73.50 | 42.02 | 44.11 |
| Qwen2-Audio | 7B | 45.03 | 84.84 | 39.84 | 81.03 | 55.99 | 92.84 | 10.31 | 14.57 | 10.36 | 11.02 | 69.42 | 45.79 | 46.75 |
| Qwen2.5-Omni | 7B | 43.49 | 84.40 | 35.86 | 82.39 | 57.02 | 93.29 | 10.52 | 13.82 | 10.49 | 12.06 | 70.17 | 45.28 | 46.57 |
| Audio Flamingo 3 | 7B | 46.43 | 85.61 | 40.04 | 83.01 | 56.47 | 93.72 | 10.11 | 14.24 | 10.38 | 10.77 | 68.97 | 47.63 | 47.28 |
| MiDaShengLM | 7B | 60.80 | 80.89 | 34.81 | 73.45 | 65.32 | 93.90 | 9.69 | 16.78 | 10.49 | 8.86 | 64.08 | 40.97 | 46.67 |
| ECHO | tiny | 78.89 | 46.60 | 54.95 | 88.96 | 71.54 | 92.81 | 9.14 | 17.77 | 14.07 | 21.26 | 69.61 | 48.64 | 51.19 |
| | small | 78.48 | 49.13 | 57.89 | 88.47 | 64.78 | 92.34 | 10.27 | 16.65 | 12.80 | 16.39 | 69.23 | 48.63 | 50.42 |
| FISHER | tiny | 80.66 | 77.88 | **70.92** | 87.85 | 67.58 | 96.02 | 12.91 | **19.71** | 17.94 | 23.07 | 68.19 | **53.08** | 56.33 |
| | mini | 81.32 | 78.06 | 70.75 | 88.85 | 65.53 | 95.19 | 14.85 | 17.92 | **19.07** | **23.81** | 68.51 | 52.22 | 56.34 |
| | small | **84.59** | 80.93 | 67.77 | **91.40** | **71.72** | **96.32** | 13.55 | 16.64 | 15.53 | 20.80 | 70.76 | 51.68 | **56.81** |

Different colors denote speech encoders, audio encoders, music encoders, LALM audio encoders, other encoders and our model respectively.

ing evaluation, we first extract the frame-level embeddings from the last layer and mean-pool them along the time axis to form the utterance-level embeddings.

### A.3.2 AUDIO ENCODERS

We compare several top audio encoders: PaSST, AudioMAE, BEATs, OpenBEATs, EAT, CED, and DaSheng, whose pre-training scenarios are more aligned with the RMIS Benchmark.

For PaSST (Koutini et al., 2022), we utilize the official passt-s-f128-p16-s10-ap.476-swa checkpoint. We use the internal mel-spectrogram parameters and apply mean-pooling over the sequence dimension to obtain the representation. As the model is pre-trained on 32 kHz audio, it is the only model in our evaluation that takes 32 kHz input, serving to demonstrate that a higher sampling rate brings limited performance gain.

For AudioMAE (Huang et al., 2022), we utilize the official base checkpoint. During evaluation, we extract the patch embeddings from the last ViT layer of the encoder and take the mean along the sequence to form the signal representation.

For BEATs (Chen et al., 2023b), we utilize the official iter3 checkpoint, which outperforms the iter1, iter2 and iter3+ checkpoints on the RMIS benchmark. The representation procedure is identical with AudioMAE.

For OpenBEATs (Bharadwaj et al., 2025), we utilize the official base and large checkpoints from the latest publicly released iter3 version. Since the model architecture is identical to BEATs, all other procedures follow those used for BEATs.

For EAT (Chen et al., 2024), we utilize the official base30 and large checkpoints. During evaluation, we take the [CLS] embedding as the signal representation.

For CED (Dinkel et al., 2024a), we utilize checkpoints of all four scales, which is similar with the scale hierarchy of FISHER. Therefore, CED is most favored for comparison to reveal the superiority of the pre-training scheme. The representation procedure is also identical to AudioMAE.

For DaSheng (Dinkel et al., 2024b), we utilize the official base, 0.6B and 1.2B checkpoints to evaluate the performances of models with much larger size. During evaluation, we extract the frame-level embeddings from the last layer and mean-pool them to retrieve the signal representation.

### A.3.3   MUSIC ENCODERS

Recent years saw specialized pre-trained models for music data, thus we also incorporate a music encoder in the comparison: MuQ (Zhu et al., 2025). We utilize the official msd checkpoint. We obtain the frame-level embeddings of the signal following the official procedure, and mean-pool them to form the utterance-level embeddings. As the only baseline pre-trained exclusively on large-scale music data, it serves as a domain-specific baseline for comparison.

### A.3.4   LARGE AUDIO LANGUAGE MODELS

We are curious about 1) how well current LALMs can understand and analyze industrial signals 2) can text-audio supervised fine-tuning (SFT) further improve the model performance on signal analysis. As pointed out in the MMAR benchmark (Ma et al., 2025), the capability of the audio encoder in the LALM is the bottleneck for advanced audio understanding. Therefore, we also evaluate the audio encoders of SOTA LALMs, including Qwen2-Audio, Qwen2.5-Omni, Audio Flamingo 3, and MiDaShengLM. For these models, we employ the official pre-trained checkpoints without task-specific fine-tuning and extract the audio encoders for evaluation.

For Qwen2-Audio (Xu et al., 2025), the audio encoder is initialized from Whisper-large-v3 and further pre-trained on large-scale audio-text pairs. We extract embeddings from the final layer of the encoder and apply mean-pooling over the sequence dimension to obtain the signal representation.

For Qwen2.5-Omni (Xu et al., 2025), the audio encoder is initialized from Whisper-large-v3 and further pre-trained in a multi-modal setting. We extract embeddings from the final layer of the encoder and apply mean-pooling over the sequence dimension to obtain the signal representation.

For Audio Flamingo 3 (Goel et al., 2025), the audio encoder is a unified AF-Whisper encoder initialized from Whisper-large-v3 and further pre-trained for long-form audio understanding. We extract embeddings from the final layer of the encoder and apply mean-pooling over the sequence dimension to obtain the signal representation.

For MiDaShengLM (Dinkel et al., 2025), the audio encoder shares the same architecture as DaSheng-0.6B but with different weights. The feature extraction procedure is identical to that of vanilla DaSheng. As it is further trained on text-audio pairs, we feel it necessary to investigate whether fine-tuning on text-audio pairs can further improve the model performance on the RMIS benchmark.

### A.3.5   OTHER ENCODERS

ECHO (Zhang et al., 2025b) is a follow-up work of FISHER with mostly identical designs, while it additionally injects frequency-aware information into the positional encoding. Therefore, we feel it necessary to add it into comparison. Specifically, we evaluate the official tiny and small checkpoints (the only sizes publicly released). We extract signal representations by concatenating features from multiple frequency bands, following the author-provided procedure.

## A.4 Efficient Scaling on Signal Analysis

In this paper, we also investigate how to effectively scale on signal analysis tasks, which has rarely been discussed in previous works. As known, the scaling law (Kaplan et al., 2020) emphasizes key factors for developing large AI models, namely model size, data volume, computational resource and test time. By scaling these factors up, the model performance will grow steadily and is predictable. While model size, data volume and computational resource must be scaled up cohesively in pre-training, test-time scaling (TTS) (Zhang et al., 2025a) focuses on adaptively adjusting test time resources and strategies for better performance.

It is worthy to note that directly scaling up pre-training has encountered bottlenecks on signal analysis. For all baselines, the performance grows steadily as the model size scales from tiny to base (around 100M), yet unexpectedly drops as the scaling continues, which seems to contradict the scaling law. As a comparison, on general audio understanding benchmarks such as Audioset (Gemmeke et al., 2017), HEAR (Turian et al., 2022) and MMAR (Ma et al., 2025), the performance can further improve as the model size scales up from 100M to billions. We believe that this is due to the poor quality of signal data for pre-training large-scale models. Industrial signals are sometimes extremely stationary and invariant. Despite the seemingly ample volume of signal data incorporated in the pre-training dataset, these data exhibit a high level of similarity and only a small amount of data remains after deduplication. These efficient data are only sufficient for training models of limited scale, and the inflection point is probably situated around 100M. If the model size is further increased, the model will be overfitted on other types of data, causing the performance on signal representation tasks to drop gradually.

Therefore, greater emphasis should be accorded to data preparation when scaling up the model. The training data should be enlarged with more unique and non-duplicated signals, such as signals of new modalities, new machinery, new working conditions, etc. Thus, it is necessary to carry out data cleaning on a broader scale and with finer granularity. Meanwhile, due to the high tendency of signal repetition, the proportion of data volume in the training configuration of the signal model should be larger than that of the speech model. As an example, FISHER-tiny trained with excessive data, performs unexpectedly well on the RMIS benchmark.

On the other hand, TTS appears to be effective on the RMIS benchmark. As an example, FISHER utilizes extra resources for inferring signals with higher sampling rates, which can be regarded as a form of TTS. We believe signal analysis requires deep thinking and long reasoning, just like the way how skilled workers take time to diagnose the malfunction. Therefore, TTS is believed to be a potential breakthrough point.

## A.5 Use of LLMs

We use LLMs to assist in related work retrieval, code writing and paper polishing.

