# OpenReview forum: "FISHER: A Foundation Model for Multi-Modal Industrial Signal Comprehensive Representation"
_ICLR.cc/2026/Conference — ICLR 2026 Conference Withdrawn Submission_

### Official Review · Reviewer_LGGf · 2025-10-28

**Soundness:** 3
**Presentation:** 3
**Contribution:** 2
**Rating:** 4
**Confidence:** 4

**Summary:**

This paper proposes FISHER, a foundation model for multi-modal industrial signal representation, aiming to address the heterogeneity of industrial data, which the authors summarize as the M5 problem. The authors also introduce a new benchmark, RMIS, to evaluate general representation ability via KNN inference without fine-tuning.

**Strengths:**

1.	Industrial signal modeling is an underexplored domain compared to speech/audio; addressing the M5 heterogeneity problem with a scalable foundation model is meaningful.
2.	The idea of modeling higher sampling rates as concatenated sub-band information is simple yet effective.
3.	The proposed RMIS benchmark covers diverse tasks and modalities, potentially serving as a valuable testbed for future research.

**Weaknesses:**

1.	The core architecture and SSL setup (teacher-student distillation via EMA, ViT backbone) are nearly identical to existing work, raising concerns that the novelty lies mainly in the data preprocessing stage.
2.	The novelty of the approach is limited, since the method essentially reweights gradients in a way similar to prior balancing strategies, without a clearly distinguished contribution.
3.	Figures could better illustrate sub-band composition to enhance interpretability.

**Questions:**

1.	It seems that the M5 Industrial Signals are independently fed into the model, with each modality processed and trained separately, without cross-modal interaction during pretraining. Could the model potentially benefit from frequency-aware attention or cross-band interaction mechanisms, rather than pure concatenation?
2.	How are sub-band boundaries handled? Is there any information leakage or overlap between adjacent sub-bands?
3.	Does using a fixed STFT window and hop size across signals with different sampling rates risk under-resolving low-rate signals or over-smoothing high-rate ones?
4.	How was the $f_{base}$ (bandwidth unit) chosen, and how sensitive is the performance to this choice?
5.	Why was KNN inference used exclusively instead of fine-tuning or linear probing?

---

> ### Author Response · Authors · 2025-11-21
> **Response to Reviewer LGGf (1/4): Novelty**
>
> We appreciate the reviewer for recognizing our contributions in FISHER and RMIS. We would like to first clarify our main novelty and respond to the reviewer's question.
>
> > The core architecture and SSL setup (teacher-student distillation via EMA, ViT backbone) are nearly identical to existing work, raising concerns that the novelty lies mainly in the data preprocessing stage.
>
> We are studying a **brand-new domain with no prior experience**. To the best of our knowledge, we are the first to explore modeling M5 industrial signals all in one model. Therefore, our novelty can be summarized as: **We are the first to demonstrate the feasibility of training a unified foundation model for industrial signals that can generalize on multiple signal analysis tasks across modalities. We provide concrete and valuable experiences for this new domain, which are listed as follows:**
>
> 1. **M5 Problem and Roadmap**: We identify and formulate the key challenges as the M5 problem, and provide a prioritized roadmap for solving it:
>    1. Special model design for active adaptation to multi-sampling-rate.
>    2. Large-scale pre-training to smooth multi-modal, multi-scale and multi-task.
>    3. High-quality representation for analyzing minim fault (anomaly detection).
>
> 2. **Multi-sampling-rate Pre-training**: Never has multi-sampling-rate pre-training been discussed in previous works. We propose an intuitive and effective sub-band modeling method based on Fourier Transform, which enables the model to adaptively process the native sampling rate without resampling.
>
> 3. **Critical Importance of Native Sampling Rate**. Our FISHER model excels among top encoders on the RMIS benchmark, suggesting that leveraging native high sampling rate is crucial for high-quality signal representation, which is a finding that has been ignored in other domains.
>
> 4. **Data Curation Insights**. We provide novel, critical and empirical guidelines for pre-training data selection in this domain, which are non-trivial:
>    1. Should use high-quality data from other domains that have similar inductive bias with industrial signals.
>    2. **Audio is most effective. Music is also beneficial. Speech is somehow interfering due to unmatched inductive bias.**
>    3. Directly adding industrial signals in pre-training data deprecates the performance. We believe this is due to the high repeatability of industrial signals, requiring specialized cleaning and curation procedures.
>
> 5. **Benchmark for Community**. We contribute the first benchmark for industrial signal representation evaluation, i.e. RMIS, to facilitate future research in this nascent field.
>
> We appreciate the reviewer's focus on novelty. However, we would like to clarify that our primary objective is to verify the feasibility of industrial signal unified modeling, and **the SSL scheme is only a part of it**. To ensure **our conclusion is not confounded by training instabilities**, we carefully select a well-established teacher-student SSL scheme that has been verified in computer vision (DINO series) and audio processing (D2V, EAT).
>
> In fact, we did explore several novel modifications tailored to signal characteristics on the SSL framework, such as [CLS] fusion and teacher decoder, whose results are presented downwards. However, none of these modifications is as satisfactory as the original SSL scheme, suggesting that for this specific domain, adopting the off-the-shelf modules provides essential stability. Further improvements shall require more specialized, signal-information-informed designs that take time to develop, which will be our future work. Our ultimate goal is to verify the feasibility of unified modeling, and to boost the performance. **We do not want to excessively pursue novelty at the expense of performance degradation.**

---

> ### Author Response · Authors · 2025-11-21
> **Response to Reviewer LGGf (2/4): Questions**
>
> > The novelty of the approach is limited, since the method essentially reweights gradients in a way similar to prior balancing strategies, without a clearly distinguished contribution.
>
> We understand this issue may refer to Equation 4, i.e. $L=L_{band}+L_{patch}$, where we combine the two sub-losses with equal weights, rather than weighted sum. We would like to clarify that equal weighting has been demonstrated in many previous works like DINO v3 (CV) and EAT (Audio) to be the most effective way for the model to equally learn global semantics and local features.
>
> > Figures could better illustrate sub-band composition to enhance interpretability.
>
> We thank the reviewer for the feedback. We have added a figure that focuses on the details of band splitting and masking strategy.
>
> > Q1: It seems that the M5 Industrial Signals are independently fed into the model, with each modality processed and trained separately, without cross-modal interaction during pretraining. Could the model potentially benefit from frequency-aware attention or cross-band interaction mechanisms, rather than pure concatenation?
>
> We thank the reviewer for the valuable suggestion. In fact, we did try out some novel band fusion methods, both in the pre-training stage and in the evaluation stage. However, none of these modifications is as robust as simple concatenation. We briefly introduce these modifications as follows.
>
> In the teacher-student self-distillation framework, the teacher has access to the entire input, while the student only has access to a portion of it. The core of this framework is how to leverage the information gap between the two models. In the scheme of band splitting, this discrepancy is further strengthened: the teacher sees the entire STFT, while the student only sees the unmasked part of the sub-band. Therefore, we have tried to modify the distillation objective to comply with the enlarged discrepancy.
>
> - [CLS] Fusion. For each sub-band, the teacher sub-band representation $t_{band}$ is the combination of all sub-band [CLS] outputs. We experiment with mean pooling, weighted pooling (the closer two sub-bands are, the higher the weight is) and self-attention.
>
> - Teacher Decoder. We introduce an additional teacher decoder to smooth the patch features of all sub-bands. The teacher decoder can be either trainable or an EMA version of the student encoder. Therefore, the teacher patch representation $t_{patch}$ can sense long-range dependencies.
>
> | Goal | Variant | Anomaly Detection | Fault Diagnosis | RMIS |
> |:-----|:------:|:------:|:------:|:------:|
> | [CLS] Fusion | mean pool | 59.66 | 58.71 | 59.17 |
> | [CLS] Fusion | weighted pool | 60.74 | 63.20 | 61.97 |
> | [CLS] Fusion | self-attention | fail | fail | fail |
> | Teacher Decoder | EMA | 56.41 | 56.33 | 56.37 |
> | Teacher Decoder | learnable | fail | fail | fail |
> | FISHER-small | None | **61.15** | **63.31** | **62.23** |
>
> The experiment results are presented upwards. [CLS] fusion with self-attention and learnable teacher decoder all fail to converge during pre-training. Meanwhile, three other modifications are considerably lower than the original one. These results strongly suggest that for this specific domain, adopting the off-the-shelf framework provides essential stability for verifying the feasibility of unified modeling. Further improvement shall require more specialized, signal-information-informed designs that take time to develop.

---

> ### Author Response · Authors · 2025-11-21
> **Response to Reviewer LGGf (3/4): Questions**
>
> > Q2: How are sub-band boundaries handled? Is there any information leakage or overlap between adjacent sub-bands?
>
> In FISHER, sub-bands are split with fixed bandwidth $w$ without overlapping. During pre-training, $sr_{batch}$ must be a multiple of $f_{base}$, thus all STFT spectrograms can be precisely divided into sub-bands without residual. During inference, since the sampling rate is varied, it is inevitable to have residuals after band splitting. Therefore, $f_{base}$ is carefully selected to minimize the residual. We will discuss it in detail in the response to Q4.
>
> According to information theory, information loss is inevitable during the band splitting process. However, we would like to emphasize that **what matters is how well the model can grasp the input information, rather than how much information is sent into the model.** During the development of FISHER, we find that band splitting creates information fragments that are easier for the model to understand, and feature concatenation is indeed an intuitive and informative way that prompts the model to combine the information of different sub-bands. Though with information leakage, the sub-band modeling is more effective for learning signals with varied sampling rates.
>
> In concern of the residual caused by band splitting, we also experiment with different strategies for handling the residual of band splitting during inference.
>
> - Drop_high (default): Start band splitting from low frequency, and drop high frequency residuals.
> - Drop_low: Start band splitting from high frequency, and drop low frequency residuals.
> - Pad_high: Pad zeros to the high frequency so that it can be divided by bandwidth $w$.
> - Expand_high: Expand the high frequency residual to lower frequency to form an overlapped sub-band.
>
> We experiment on datasets that would yield residual in band splitting, i.e. IIEE, 2 MaFaulDa, 2 SDUST and 4 UMGED. We compare the average accuracies on these datasets. The results are presented as follows:
>
> | Scale | drop_high | drop_low | pad_high | expand_high |
> |:-----|:------:|:------:|:------:|:------:|
> | tiny | **58.10** | 56.12 | 57.59 | 57.94 |
> | mini | **58.20** | 56.49 | 58.06 | 58.04 |
> | small | 57.43 | **57.78** | 57.50 | 57.20 |
>
> Surprisingly, directly dropping high frequency residuals turns out to be the most robust strategy, suggesting that dropping residual information is more beneficial for the overall representation.
>
>
> > Q3: Does using a fixed STFT window and hop size across signals with different sampling rates risk under-resolving low-rate signals or over-smoothing high-rate ones?
>
> We appreciate the reviewer for identifying this critical issue. We would like to clarify that this issue is dealt by our novel fixed-time-duration design theoretically. In FISHER, we adopt fixed-time-duration STFT window $t_{win}=25\ \text{ms}$ and hop size $t_{hop}=10\ \text{ms}$. Therefore, the temporal and frequency resolutions are fixed to the same physical measure for arbitrary sampling rates.
>
> - Temporal Resolution: ${\Delta}t=t_{hop}=10\ \text{ms}$
> - Frequency Resolution: ${\Delta}f=\frac{1}{t_{win}}=40\ \text{Hz}$ (Proof can be found in Equation (1))
>
> Here the resolution is defined as the gap between adjacent time / frequency indices of the STFT. Therefore, we are analyzing all signals with the identical scale, thereby preventing both under-resolving and over-smoothing.

---

> ### Author Response · Authors · 2025-11-21
> **Response to Reviewer LGGf (4/4): Questions**
>
> > Q4: How was the $f_{base}$ (bandwidth unit) chosen, and how sensitive is the performance to this choice?
>
> Common sampling rates are: 12.8 kHz, 16 kHz, 25.6 kHz, 32 kHz, 44.1 kHz, 48 kHz, 51.2 kHz, 64 kHz. According to the Nyquist Sampling Theorem, the Nyquist bandwidth is half the sampling rate, i.e. 6.4 kHz, 8 kHz, 12.8 kHz, 16 kHz, 22.05 kHz, 24 kHz, 25.6 kHz, 32 kHz. Therefore, $f_{base}$ is chosen based on four principles:
>
> - Should be or close to a common factor of as many Nyquist bandwidths as possible.
> - Should be lower than the minimum Nyquist bandwidth.
> - Should be higher than the frequency range of a patch (1.28 kHz for 16\*16 patch).
> - Should be or slightly bigger than a multiple of the frequency range of a patch to reduce information loss during patchify.
>
> Based on these principles, we filter out two proper choices for $f_{base}$, i.e. 2 kHz and 4 kHz. We experiment with these two $f_{base}$ on three different scales. The results are depicted as follows:
>
> | Scale | $f_{base}$ | $w$ | Anomaly Detection | Fault Diagnosis | RMIS |
> |:-----|:------:|:------:|:------:|:------:|:------:|
> | tiny | 2 kHz | 50 | 58.87 | 62.52 | 60.70 |
> | tiny | 4 kHz | 100 | **59.11** | **63.13** | **61.12** |
> | mini | 2 kHz | 50 | **59.85** | 62.72 | 61.28 |
> | mini | 4 kHz | 100 | 59.60 | **63.25** | **61.43** |
> | small | 2 kHz | 50 | **61.15** | **63.31** | **62.23** |
> | small | 4 kHz | 100 | 59.44 | 63.24 | 61.34 |
>
> It suggests that the tiny and mini versions favor larger input bandwidth, while the small version favors smaller input bandwidth. Though with fluctuation, the RMIS scores of all eight checkpoints are significantly higher than top baselines such as BEATs (57.30), EAT-large (58.03) and ECHO-small (59.00), suggesting that the band-splitting scheme is pretty robust to $f_{base}$.
>
> > Q5: Why was KNN inference used exclusively instead of fine-tuning or linear probing?
>
> We believe KNN inference is the most suitable metric for our topic due to the following reasons:
>
> 1. We focus on industrial signal foundation model, which emphasizes the capability to generalize on various analysis tasks without fine-tuning. The industries are in great need of such kind of model so that different signals can be processed by the same model.
>
> 2. Linear probing can not be applied to anomaly detection tasks, since anomalies are detected by KNN inference. Details of the anomaly detection procedure can be found in Section A.1.1. Full fine-tuning is applicable, yet it is tedious and inefficient, since it has to be run under multiple seeds to reduce fluctuation (ASD results commonly fluctuate more than 1\%).
>
> 3. Some fault diagnosis datasets are extremely easy under linear probing and full fine-tuning settings, making it hard to compare the actual model capabilities.
>
> To evaluate the encoder more thoroughly, we extend the criteria to both full fine-tuning and linear probing as suggested. The experiments are still running, and we are making efforts to ensure that the evaluation is fair and consistent for all encoders. The results will be added in the paper once all the experiments are finished. We provide preliminary results on ASD full fine-tuning:
>
> | Model | Variant | Size | 20 | 21 | 22 | 23 | 24 | 25 | Mean |
> |:-----|:----:|:---:|:---:|:---:|:---:|:---:|:---:|:---:|:---:|
> | BEATs | iter3 | 90M | **91.97** | **70.90** | 65.96 | 70.05 | **67.12** | **61.93** | **71.32** |
> | EAT | base30 | 88M | 89.51 | 69.47 | 65.17 | 69.28 | 64.95 | 60.89 | 69.88 |
> | FISHER | mini | 10M | 88.44 | 67.10 | 65.55 | 69.35 | 62.98 | 60.07 | 68.92 |
> | FISHER | small | 22M | 88.54 | 68.42 | **66.53** | **72.94** | 64.39 | 61.44 | 70.38 |
>
> We adopt the SOTA fine-tuning method in [1, 2], which fine-tunes the model on all ASD datasets with learnable sub-centers for machines without annotated working conditions. We fine-tune each model under 5 different seeds and report the arithmetic average.
>
> We compare FISHER with the best encoders for ASD, i.e. BEATs and EAT, where BEATs > FISHER-small > EAT-base30, with the same order as KNN inference. However, it is noted that the gap between BEATs and FISHER is slightly shortened from 1.01\% to 0.94\% after switching to full fine-tuning, while the gap between FISHER and EAT is slightly widen from 0.48\% to 0.5\%. It suggests that FISHER is also suitable for downstream fine-tuning.
>
> [1] Jiang A, Liang W, Feng S, et al. THUEE SYSTEM FOR DCASE 2025 ANOMALOUS SOUND DETECTION CHALLENGE[R]. DCASE2025 Challenge, Tech. Rep, 2025.
>
> [2] Zheng X, Jiang A, Han B, et al. SJTU-AITHU system for DCASE 2025 anomalous sound detection challenge[R]. DCASE2025 Challenge, Tech. Rep, 2025.

---

### Official Review · Reviewer_41iu · 2025-10-31

**Soundness:** 3
**Presentation:** 2
**Contribution:** 2
**Rating:** 4
**Confidence:** 2

**Summary:**

This paper presents FISHER, a new foundation model aimed at creating a unified representation for industrial signals, a domain the authors characterize with the "M5 problem." The central and most compelling idea is to reframe the challenge of variable sampling rates by modeling signals as a composition of frequency sub-bands. This allows for a consistent processing architecture, which is trained via a teacher-student self-distillation scheme. To properly assess the model's capabilities, the authors have also constructed a new and comprehensive benchmark, RMIS. The experimental evaluation is strong.

**Strengths:**

The authors should be commended for tackling a problem of significant practical importance. Creating a single model for the diverse and heterogeneous world of industrial signals is a valuable goal, and this work makes a convincing step in that direction. The key strength of the paper, in my view, is the conceptual novelty of the sub-band approach. It's an elegant and intuitive way to handle the variable-length nature of signals sampled at different rates. This strong methodological contribution is backed by a very thorough empirical evaluation. The scale of the comparison, across the new RMIS benchmark and against 15 different encoders, lends significant weight to the authors' claims. Furthermore, the development of the RMIS benchmark itself stands out as a valuable contribution that will surely benefit the wider research community.

**Weaknesses:**

While the paper is promising, its clarity could be significantly improved in several areas, which currently hinders a full appreciation of the work. My primary concern is that the paper is not fully self-contained. For example, a key component of the training, the "mask cloning strategy," is mentioned without explanation, forcing readers to consult the external EAT paper to understand the methodology. Similarly, the crucial process of how signals yielding a variable number of sub-bands are actually batched together is left ambiguous. Beyond these clarity issues, some key design choices lack justification. It's unclear why the distillation loss is computed on unmasked patches, a deviation from common practice that warrants an explanation. The paper would also benefit from a deeper analysis of its own results. The strong performance of the BEATs model on anomaly detection is a fascinating finding, and the lack of discussion around why this might be the case is a missed opportunity for richer scientific insight.

**Questions:**

To help clarify some of the points above, I would appreciate your response to the following:
1. Could you please elaborate on the practical batching mechanism for inputs that have different sampling rates and thus produce a variable number of sub-bands?
2. Regarding the training process, what was the specific motivation for including the unmasked patches in the distillation loss, and what effect did you observe from this choice?
3. To make the paper more self-contained, would it be possible to provide a brief, high-level explanation of the "mask cloning strategy" and its specific role in your framework?
4. I would be very interested to hear your thoughts on the exceptional performance of the BEATs model in the anomaly detection tasks. Do you have a hypothesis for this result, and what might it suggest about the nature of those specific tasks?

---

> ### Author Response · Authors · 2025-11-21
> **Response to Reviewer 41iu (1/2)**
>
> We are glad the reviewer commended our novelty and contributions. We also greatly appreciate the reviewer for pointing out some unclear concepts. We have revised the paper to make these concepts intuitive and self-contained. We have also added ablation studies to better validate the effectiveness as suggested by other reviewers. We now respond to the reviewer's questions and describe our modifications.
>
> > Similarly, the crucial process of how signals yielding a variable number of sub-bands are actually batched together is left ambiguous.
> >
> > Could you please elaborate on the practical batching mechanism for inputs that have different sampling rates and thus produce a variable number of sub-bands?
>
> We are glad to explain the batching mechanism in detail. During pre-training, for a batch of signals with varied sampling rates, we first resample them to a fixed sampling rate $sr_{batch}$, which is randomly selected from all the harmonics of $f_{base}$ that are smaller than a maximum frequency $f_{max}$. That is to say, all samples of a batch will have the same shape `(T, F)`, while $sr_{batch}$ changes every 50 steps (50 * num_gpu batches). In that way, all samples will have the same number of sub-bands, and the number of sub-bands will also be the same across GPUs, thereby **preventing load imbalance in multi-GPU training**. We then calculate the STFT of these resampled signals, resulting in shape `(B, T, F)`. Then we apply band splitting with fixed bandwidth $w$, resulting in shape `(B, T, w, n)`. Then we rearrange the tensor to `(n, B, T, w)`, and merge the first two axes together, resulting in shape `(B*n, T, w)`. Therefore, what the model actually sees is a batch of sub-bands.
>
> During inference, we pre-split signals with varied sampling rates into sub-bands with fixed bandwidth $w$, resulting in `(n_i, T, w)` where n_i is varied for different samples. We put these sub-bands into a first-in-first-out queue. The collator then reads the queue for m sub-bands a time and batches these sub-bands, where m is determined by the computation resource. The model processes these sub-bands individually, and we concatenate the corresponding sub-bands afterwards to form the signal representation.
>
> > Beyond these clarity issues, some key design choices lack justification. It's unclear why the distillation loss is computed on unmasked patches, a deviation from common practice that warrants an explanation.
> >
> > Regarding the training process, what was the specific motivation for including the unmasked patches in the distillation loss, and what effect did you observe from this choice?
>
> We sincerely apologize for the error in Equation (3). The patch level loss is actually computed only on masked patches. We reformulate the patch loss part in Equation (3) as:
>
> $$L_{patch}=\|(s_{patch}^{(mask)}-sg(t_{patch}^{(mask)})\|_2^2$$
>
> where the $(mask)$ superscript selects all masked patches, and $sg(\cdot)$ denotes stop gradient.
>
> We agree that the masked part is much more effective for pre-training and we greatly appreciate the reviewer for spotting this anomaly.

---

> ### Author Response · Authors · 2025-11-21
> **Response to Reviewer 41iu (2/2)**
>
> > For example, a key component of the training, the "mask cloning strategy," is mentioned without explanation, forcing readers to consult the external EAT paper to understand the methodology.
> >
> > To make the paper more self-contained, would it be possible to provide a brief, high-level explanation of the "mask cloning strategy" and its specific role in your framework?
>
> We appreciate the reviewer for pointing out this important concept that requires further clarification. We have revised Section 3.2 to better explain how mask cloning is conducted.
>
> We employ mask cloning in the masking stage of the student encoder. After splitting a clip STFT into $n$ sub-bands, we apply multiple masks to each of these sub-bands to generate multiple views of the same sub-band. We constrain the maximum number of views sourced from the same clip as $m_b$ to prevent GPU load imbalance. These views can reuse the same teacher representation, and the teacher only has to forward pass once to supervise $\lfloor\frac{m_b}{n}\rfloor$ views.
>
> Applying mask cloning equivalently increases the batch size with less computation and I/O overhead. This is especially helpful for FISHER since processing a sub-band is much faster than processing a complete spectrogram, thus requiring much faster data reading. We encountered severe I/O bottlenecks with low $m_b$ when developing FISHER.
>
> > The strong performance of the BEATs model on anomaly detection is a fascinating finding, and the lack of discussion around why this might be the case is a missed opportunity for richer scientific insight.
> >
> > I would be very interested to hear your thoughts on the exceptional performance of the BEATs model in the anomaly detection tasks. Do you have a hypothesis for this result, and what might it suggest about the nature of those specific tasks?
>
> This is an insightful question, and we are more than willing to discuss it. As known, the pre-training procedure of BEATs requires explicitly tokenizing patches into discrete token ids, while other models are pre-trained in continuous manners. In that way, the patch features of BEATs may resemble some kind of codewords for an underlying machine language, probably related to some discrete hidden states of the machine (like the finite-state machine model). The ASD task happens to suit such a kind of modeling scheme. As a result, BEATs is generally superior on ASD tasks.
>
> However, this scheme does not apply well to fault diagnosis tasks. The fault diagnosis score of BEATs is notably lower than the scores of EAT and FISHER, and the same is true for the overall RMIS score. Moreover, the training cost of BEATs is significantly higher, since it needs to iterate between prediction and tokenization four times. Therefore, we build our work upon the EAT paradigm for better performance and efficiency.

---

### Official Review · Reviewer_xTTZ · 2025-11-01

**Soundness:** 3
**Presentation:** 3
**Contribution:** 2
**Rating:** 4
**Confidence:** 3

**Summary:**

The paper proposed the M5 problem, and presents FISHER, a foundation model that represents signals as STFT sub-bands, treats increments in sampling rate as concatenation of additional sub-band information, and uses a teacher–student self-supervised pretraining framework. To evaluate, the authors introduce RMIS benchmark spanning 19 datasets across four modalities. FISHER outperforms 15 state-of-the-art encoders by at least 3.23% on average, showing efficient scaling curves.

**Strengths:**

- Good problem formulation: clearly formalizing the M5 problem is a valuable contribution.

**Weaknesses:**

- Limited novelty: many modules in this paper can be found in references. For example, ViT, features emerging, and EMA are off-the-shelf modules. It’ll be great if the authors could explain more about what’s the unique contribution and novelty in this paper.

**Questions:**

- What’s the model sizes of baselines? In Appendix, authors described some but not all model sizes of baselines. To better understand the efficient scaling curves of FISHER, it'll be great if authors could display the number of parameters for all baselines.
 - Should we train FISHER with pretrained weights instead of training from scratch? In AST (https://arxiv.org/abs/2104.01778), with the pretrained ViT weights, the performance of AST is further boosted.

---

> ### Author Response · Authors · 2025-11-21
> **Response to Reviewer xTTZ (1/2): Novelty**
>
> We are glad that the reviewer recognized the formulation of the M5 problem. We would like to first address the reviewer's concern in novelty, and respond to detailed questions.
>
> > Limited novelty: many modules in this paper can be found in references. For example, ViT, features emerging, and EMA are off-the-shelf modules. It’ll be great if the authors could explain more about what’s the unique contribution and novelty in this paper.
>
> We are studying a **brand-new domain with no prior experience**. To the best of our knowledge, we are the first to explore modeling M5 industrial signals all in one model. Therefore, our novelty can be summarized as: **We are the first to demonstrate the feasibility of training a unified foundation model for industrial signals that can generalize on multiple signal analysis tasks across modalities. We provide concrete and valuable experiences for this new domain, which are listed as follows:**
>
> 1. **M5 Problem and Roadmap**: We identify and formulate the key challenges as the M5 problem, and provide a prioritized roadmap for solving it:
>    1. Special model design for active adaptation to multi-sampling-rate.
>    2. Large-scale pre-training to smooth multi-modal, multi-scale and multi-task.
>    3. High-quality representation for analyzing minim fault (anomaly detection).
>
> 2. **Multi-sampling-rate Pre-training**: Never has multi-sampling-rate pre-training been discussed in previous works. We propose an intuitive and effective sub-band modeling method based on Fourier Transform, which enables the model to adaptively process the native sampling rate without resampling.
>
> 3. **Critical Importance of Native Sampling Rate**. Our FISHER model excels among top encoders on the RMIS benchmark, suggesting that leveraging native high sampling rate is crucial for high-quality signal representation, which is a finding that has been ignored in other domains.
>
> 4. **Data Curation Insights**. We provide novel, critical and empirical guidelines for pre-training data selection in this domain, which are non-trivial:
>    1. Should use high-quality data from other domains that have similar inductive bias with industrial signals.
>    2. **Audio is most effective. Music is also beneficial. Speech is somehow interfering due to unmatched inductive bias.**
>    3. Directly adding industrial signals in pre-training data deprecates the performance. We believe this is due to the high repeatability of industrial signals, requiring specialized cleaning and curation procedures.
>
> 5. **Benchmark for Community**. We contribute the first benchmark for industrial signal representation evaluation, i.e. RMIS, to facilitate future research in this nascent field.
>
> We appreciate the reviewer's focus on novelty. However, we would like to clarify that our primary objective is to verify the feasibility of industrial signal unified modeling, and **the SSL scheme is only a part of it**. To ensure **our conclusion is not confounded by training instabilities**, we carefully select a well-established teacher-student SSL scheme that has been verified in computer vision (DINO series) and audio processing (D2V, EAT).
>
> In fact, we did explore several novel modifications tailored to signal characteristics on the SSL framework, such as [CLS] fusion and teacher decoder. The methodology and result can be found in Part 2 of the response to Reviewer LGGf. However, none of these modifications produce satisfactory results as the original SSL scheme, suggesting that for this specific domain, adopting the off-the-shelf modules provides essential stability. Further improvements shall require more specialized, signal-information-informed designs that take time to develop, which will be our future work. Our ultimate goal is to verify the feasibility of unified modeling, and to boost the performance. **We do not want to excessively pursue novelty at the expense of performance degradation.**
>
> As a summary, FISHER **demonstrates the feasibility of unified signal modeling and provides valuable insights on the importance of native sampling rate and data selection**. The performance of FISHER scales efficiently on the RMIS benchmark compared to other SOTA encoders. Thus, we believe that convincingly establishing this foundation and providing a benchmark for the community is a significant contribution in itself, and we hope the reviewer can recognize its value.

---

> ### Author Response · Authors · 2025-11-21
> **Response to Reviewer xTTZ (2/2): Questions**
>
> > What’s the model sizes of baselines? In Appendix, authors described some but not all model sizes of baselines. To better understand the efficient scaling curves of FISHER, it'll be great if authors could display the number of parameters for all baselines.
>
> We thank the reviewer for this valuable suggestion. We have added the numerical sizes in Table 3, Table 5 and Appendix A.3.1.
>
> - Speech Encoder
>   - Wav2Vec 2.0: base 95M; large 315M; 1B 962M; 2B 2.16B
>   - Whisper: tiny 7.6M; base 19.8M; small 87M; medium 306M; large-v3 635M
> - Audio Encoder
>   - PaSST: 86M
>   - AudioMAE: 85M
>   - BEATs: 90M
>   - OpenBEATs: base 90M; large 310M
>   - EAT: base30 88M; large 310M
>   - CED: tiny 5.5M; mini: 9.6M; small: 22M; base: 86M
>   - DaSheng: base 85M; 0.6B 630M; 1.2B 1130M
> - Music Encoder
>   - MuQ: 333M
> - LALM Audio Encoder
>   - Qwen2-Audio: 637M
>   - Qwen2.5-Omni: 640M
>   - Audio Flamingo 3: 637M
>   - MiDaShengLM: 630M
> - Other Encoder
>   - ECHO: tiny 5.5M; small 22M
>
> > Should we train FISHER with pretrained weights instead of training from scratch? In AST (https://arxiv.org/abs/2104.01778), with the pretrained ViT weights, the performance of AST is further boosted.
>
> AST is an early work on audio pre-trained models. It is initialized from an image pre-trained ViT as a moderate yet unmatched starting point. However, its successor, SSAST, and recent SOTA audio pre-trained models such as BEATs, EAT, CED and DaSheng, conformably opt to pre-train from scratch. It suggests that although starting from a similar yet unmatched checkpoint does help, pre-training from scratch further improves the model's capability.
>
> As for FISHER, the input feature is switched to STFT, instead of the commonly-adopted Mel spectrogram. Therefore, open-sourced pre-trained weights are also unmatched for FISHER, and the results should not outperform training from scratch. To validate it, we first pre-train an EAT-small model on audioset with the official script, then use it to initialize FISHER and pre-train FISHER. The experiments are still running (20k steps out of 40k steps), and we present preliminary results here.
>
> | Model | Anomaly Detection | Fault Diagnosis | RMIS |
> |:-----|:------:|:------:|:------:|
> | EAT-small | 59.16 | 53.41 | 56.29 |
> | with EAT init (20k steps) | 61.00 | 63.80 | 62.40 |
> | train from scratch (20k steps) | **61.10** | **63.87** | **62.49** |
>
> The preliminary results suggest that initializing from pre-trained weights is unnecessary. We are monitoring the results to see if there is any improvement in the long run.

---

### Official Review · Reviewer_DsVz · 2025-11-03

**Soundness:** 2
**Presentation:** 2
**Contribution:** 2
**Rating:** 2
**Confidence:** 2

**Summary:**

The paper presents a foundation model for multi-modal signal representations including sound, vibration, voltage, current, temperature, and etc.

**Strengths:**

The paper introduces a foundation model for multi-modal signal representation. This is the first attempt to learn a multi-modal, multi-sampling-rate for various signal datasets.

**Weaknesses:**

My primary concern lies in the lack of novelty. I could not identify any clear novel contribution beyond the use of multi-modal training.

**[No Ablation Study]**
The paper does not include any ablation studies, making it difficult to understand why the proposed model outperforms others. It would be beneficial to provide ablation results for factors such as multi-modality, multi-scale datasets, model architecture, and sequence length.

**[Input Size]**
During training, FISHER uses a fixed 10-second segment, which limits the model’s usability. Could you clarify how the model handles longer input signals during evaluation? A 10-second limit seems too short for practical applications.

I'm not an expert in this topic, specifically in industrial signal, so please lower the weight of my score accordingly.

**Questions:**

.

**Details Of Ethics Concerns:**

.

---

> ### Author Response · Authors · 2025-11-21
> **Response to Reviewer DsVz (1/3): Novelty**
>
> We greatly appreciate the reviewer's constructive feedback. We would like to first address the reviewer's concern about novelty, and then provide additional ablation studies and respond to detailed questions.
>
> We are studying a **brand-new domain with no prior experience**. To the best of our knowledge, we are the first to explore modeling M5 industrial signals all in one model. Therefore, our novelty can be summarized as: **We are the first to demonstrate the feasibility of training a unified foundation model for industrial signals that can generalize on multiple signal analysis tasks across modalities. We provide concrete and valuable experiences for this new domain, which are listed as follows:**
>
> 1. **M5 Problem and Roadmap**: We identify and formulate the key challenges as the M5 problem, and provide a prioritized roadmap for solving it:
>    1. Special model design for active adaptation to multi-sampling-rate.
>    2. Large-scale pre-training to smooth multi-modal, multi-scale and multi-task.
>    3. High-quality representation for analyzing minim fault (anomaly detection).
>
> 2. **Multi-sampling-rate Pre-training**: Never has multi-sampling-rate pre-training been discussed in previous works. We propose an intuitive and effective sub-band modeling method based on Fourier Transform, which enables the model to adaptively process the native sampling rate without resampling.
>
> 3. **Critical Importance of Native Sampling Rate**. Our FISHER model excels among top encoders on the RMIS benchmark, suggesting that leveraging native high sampling rate is crucial for high-quality signal representation, which is a finding that has been ignored in other domains.
>
> 4. **Data Curation Insights**. We provide novel, critical and empirical guidelines for pre-training data selection in this domain, which are non-trivial:
>    1. Should use high-quality data from other domains that have similar inductive bias with industrial signals.
>    2. **Audio is most effective. Music is also beneficial. Speech is somehow interfering due to unmatched inductive bias.**
>    3. Directly adding industrial signals in pre-training data deprecates the performance. We believe this is due to the high repeatability of industrial signals, requiring specialized cleaning and curation procedures.
>
> 5. **Benchmark for Community**. We contribute the first benchmark for industrial signal representation evaluation, i.e. RMIS, to facilitate future research in this nascent field.
>
> We appreciate the reviewer's focus on novelty. However, we would like to clarify that our primary objective is to verify the feasibility of industrial signal unified modeling, and **the SSL scheme is only a part of it**. To ensure **our conclusion is not confounded by training instabilities**, we carefully select a well-established teacher-student SSL scheme that has been verified in computer vision (DINO series) and audio processing (D2V, EAT).
>
> In fact, we did explore several novel modifications tailored to signal characteristics on the SSL framework, such as [CLS] fusion and teacher decoder. The methodology and result can be found in Part 2 of the response to Reviewer LGGf. However, none of these modifications produce satisfactory results as the original SSL scheme, suggesting that for this specific domain, adopting the off-the-shelf modules provides essential stability. Further improvements shall require more specialized, signal-information-informed designs that take time to develop, which will be our future work. Our ultimate goal is to verify the feasibility of unified modeling, and to boost the performance. **We do not want to excessively pursue novelty at the expense of performance degradation.**
>
> As a summary, FISHER **demonstrates the feasibility of unified signal modeling and provides valuable insights on the importance of native sampling rate and data selection**. The performance of FISHER scales efficiently on the RMIS benchmark compared to other SOTA encoders. Thus, we believe that convincingly establishing this foundation and providing a benchmark for the community is a significant contribution in itself, and we hope the reviewer can recognize its value.

---

> ### Author Response · Authors · 2025-11-21
> **Response to Reviewer DsVz (2/3): Ablation Study**
>
> > [No Ablation Study] The paper does not include any ablation studies, making it difficult to understand why the proposed model outperforms others. It would be beneficial to provide ablation results for factors such as multi-modality, multi-scale datasets, model architecture, and sequence length.
>
> We thank the reviewer for pointing out this deficiency. We now present preliminary results of additional ablation studies. More results will be posted afterwards.
>
> ## Data Selection
>
> One of our key findings is about the data composition for training industrial signal foundation model. We figure out that industrial signals are actually of poor quality for pre-training due to high repeatability, while high-quality data from audio and music have similar inductive bias with signals and therefore constitute the majority of pre-training data.
>
> Since speech encoders are conformably ineffective on the RMIS benchmark, we do not further investigate the use of speech data in pre-training. We experiment with three different data sources:
> 1. Audio (12k hours in total): Audioset (6k hours), Freesound (6k hours).
> 2. Music (5k hours in total): MTG-Jamendo (4k hours), Music4all (1k hours).
> 3. Industrial Signals (5k hours in total): There is no open-sourced large-scale signal dataset longer than 1k hours. To validate the effect of adding signal data in pre-training, we first collect sound and vibration signals of five kinds of **real machines**: coal grinder, pump, wind turbine, water turbine and electric transformer. We then employ a CED model for data deduplication, creating a filtered dataset of 5k hours.
>
> | Data Type | Hours | Anomaly Detection | Fault Diagnosis | RMIS |
> |:-----|:------:|:------:|:------:|:------:|
> | Audio | 12k | 59.87 | 63.21 | 61.54 |
> | Audio, Music | 17k | **61.15** | **63.31** | **62.23** |
> | Audio, Music, Industrial Signal | 22k | 58.61 | 61.78 | 60.19 |
>
> We pre-train FISHER-small with three different data compositions. The model trained by pure audio provides a decent starting point of 61.54, while adding music data further boosts it to 62.23. However, the model performance drops significantly after adding industrial signal, suggesting that the filtered signal dataset is still of poor quality due to high duplication. Therefore, we believe that industrial signal requires special data curation designs before they can be successfully utilized in pre-training. On the other hand, audio and music are currently the best data sources for training signal foundation model.
>
> ## Fundamental Frequency
>
> We also experiment with different fundamental frequencies $f_{base}$ on three different scales. The results are presented as follows:
>
> | Scale | $f_{base}$ | $w$ | Anomaly Detection | Fault Diagnosis | RMIS |
> |:-----|:------:|:------:|:------:|:------:|:------:|
> | tiny | 2 kHz | 50 | 58.87 | 62.52 | 60.70 |
> | tiny | 4 kHz | 100 | **59.11** | **63.13** | **61.12** |
> | mini | 2 kHz | 50 | **59.85** | 62.72 | 61.28 |
> | mini | 4 kHz | 100 | 59.60 | **63.25** | **61.43** |
> | small | 2 kHz | 50 | **61.15** | **63.31** | **62.23** |
> | small | 4 kHz | 100 | 59.44 | 63.24 | 61.34 |
>
> It suggests that the tiny and mini versions favor larger input bandwidth, while the small version favors smaller input bandwidth. Though with fluctuation, the RMIS scores of all eight checkpoints are significantly higher than top baselines such as BEATs (57.30), EAT-large (58.03) and ECHO-small (59.00), suggesting that the band-splitting scheme is pretty robust to $f_{base}$.

---

> ### Author Response · Authors · 2025-11-21
> **Response to Reviewer DsVz: (3/3): Questions**
>
> We now respond to the reviewer's detailed questions.
>
> > I could not identify any clear novel contribution beyond the use of multi-modal training.
>
> We understand the reviewer's concern. Our novelties are summarized in Part 1. Here we would like to respectfully point out that FISHER is only pre-trained on the sound modal, with audio data and music data. As mentioned upwards, one of our key findings is that directly adding industrial signals in pre-training turns out to be harmful.
>
> > [Input Size] During training, FISHER uses a fixed 10-second segment, which limits the model’s usability. Could you clarify how the model handles longer input signals during evaluation? A 10-second limit seems too short for practical applications.
>
> | Dataset | Clip Length |
> |:-----|:------:|
> | Audioset | 10 s |
> | ESC-50 | 5 s |
> | SPC-2 | 1 s |
>
> | Audio Model | Input Length |
> |:-----|:------:|
> | AudioMAE | 10 s |
> | BEATs | 10 s |
> | EAT | 10 s |
> | CED | 10 s |
> | DaSheng | 10 s |
>
> | Fault Diagnosis Model | Input Size | Time Duration |
> |:-----|:------:|:-----:|
> | AET [1] | 2048 | 102 ms (20 kHz), 170 ms (12 kHz) |
> | GCAIPN [2] | 2560 | 100 ms (25.6 kHz), 200 ms (12.8 kHz) |
> | RAViT [3] | 2048 | 160 ms (12.8 kHz) |
> | MDIFN [4] | 1024 | 16 ms (64 kHz), 51.2 ms (20 kHz) |
>
> We respectfully disagree with the reviewer on this point. The input length is widely selected as (or less than) **10 s in audio datasets and audio pre-trained models**. As for fault diagnosis, the input length of prevalent models are **even smaller than 1 s (usually thousands of points)**, which can hardly learn long-range dependencies. Therefore, the input length of FISHER is sufficient for signal analysis, and is indeed a breakthrough compared with prevalent fault diagnosis models.
>
> For the ASD part of the RMIS benchmark, if an audio clip is longer than 10 s, we split it into 10 s segments with overlap (18 s into two 10 s), and process them individually. The clip-level representation is the mean of all segment-level representations. We use the clip-level representation for evaluation. For the fault diagnosis part of the RMIS benchmark, we pre-split clips longer than 10 s into 10 s segments, and evaluate the model on these segments. Detailed information can be found in Section 4.2 of the paper.
>
> In deployment, we segment a streaming signal by a 10 s sliding window, and analyze these segments individually. A 10 s input length is quite sufficient since the periods of characteristic frequencies are much smaller.
>
> We are willing to expand the input length to longer durations. However, current audio datasets lack both high quality and long duration data, making it non-trivial to expand. This will probably be our future work.
>
>
> [1] Wang G, Liu D, Cui L. Auto-embedding transformer for interpretable few-shot fault diagnosis of rolling bearings[J]. IEEE Transactions on Reliability, 2023, 73(2): 1270-1279.
>
> [2] Sun S, Xia X, Zhou H. A graph representation learning-based method for fault diagnosis of rotating machinery under time-varying speed conditions[J]. Nonlinear Dynamics, 2025: 1-27.
>
> [3] Lian Y, Wang J, Li Z, et al. Residual attention guided vision transformer with acoustic-vibration signal feature fusion for cross-domain fault diagnosis[J]. Advanced Engineering Informatics, 2025, 64: 103003.
>
> [4] Gao T, Yang J, Tang Q. A multi-source domain information fusion network for rotating machinery fault diagnosis under variable operating conditions[J]. Information Fusion, 2024, 106: 102278.

---

### Note · Authors · 2025-12-03

**Comment:**

We sincerely thank all the reviewers for their valuable time and meaningful feedback. The reviewers' concerns primarily focus on novelty, which has prompted us to rethink, reorganize and highlight our unique contributions. Meanwhile, we are glad that all reviewers recognize the effectiveness of our model without any questions about model performance. This is truly an incentive for us. We believe our work is of pioneering importance to both academia and industry, and we are willing to discuss it with the reviewers. However, we are sorry to hear that the rebuttal policy has changed, which bans further discussion and leaves us with no opportunity to explain our core innovations to the reviewers. Therefore, we have decided to withdraw.

**Withdrawal Confirmation:**

I have read and agree with the venue's withdrawal policy on behalf of myself and my co-authors.